# Rank one HCIZ at high temperature: interpolating between classical and free convolutions

Pierre Mergny[*,1,2] and Marc Potters[3]

[1]Chair of Econophysics & Complex Systems, Ecole polytechnique, 91128 Palaiseau Cedex, France

[2]LPTMS, CNRS, Univ. Paris-Sud, Université Paris-Saclay, 91405 Orsay, France

[3]Capital Fund Management, 23 rue de l'Université, 75007 Paris, France

We study the rank one Harish-Chandra-Itzykson-Zuber integral in the limit where $\frac{N\beta}{2} \to c$, called the high temperature regime and show that it can be used to construct a promising one-parameter interpolation, with parameter $c$ between the classical and the free convolution. This $c$-convolution has a simple interpretation in terms of another associated family of distribution indexed by $c$, called the Markov-Krein transform: the $c$-convolution of two distributions corresponds to the classical convolution of their Markov-Krein transforms. We derive first cumulants-moments relations, a central limit theorem, a Poisson limit theorem and shows several numerical examples of $c$-convoluted distributions.

## 1 Introduction

For a self-adjoint random matrix $\mathbf{A}$, of size $N$ with real, complex or quaternionic entries, under mild assumptions and up to a rescaling of the entries, we know from Random Matrix Theory (RMT) that the (random) spectral measure of $\mathbf{A}$ tends to a *deterministic* limiting measure $\mu_A$ in the limit $N \to \infty$ (see for example [1]). Free Probability, introduced by Voiculescu [2], allows one to compute the limiting spectral distribution denoted by $\mu_A \boxplus \mu_B$ and known as the *free convolution*, for the sum of two such random matrices $\mathbf{A}$ and $\mathbf{B}$, in this limit $N \to \infty$, where one replaces the notion of independence of classical probability by the notion of *freeness* of non-commutative algebraic probability theory. The correspondence between classical and free probability is given in Table 1.

For a measure $\mu_A$ with (compact) support $I$, the transform $\mathcal{R}_{\mu_A}(.)$ that linearizes the free convolution is the *R-transform* defined by:

$$\mathcal{R}_{\mu_A}(t) := \mathcal{G}_{\mu_A}^{(-1)}(t) - \frac{1}{t}, \tag{1}$$

---

[*]mergny.pierre@gmail.com

| Classical Probability | Free Probability |
|:---:|:---:|
| $X$ real random variable | $A$ self-adjoint operator |
| Independance | Freeness |
| $\mu_X * \mu_Y$ | $\mu_A \boxplus \mu_B$ |
| $\log \mathbb{E}_X \left[ e^{tX} \right]$ | $\mathcal{R}_{\mu_A}(t)$ |

**Table 1:** Correspondence between the classical and the free world

where $.^{(-1)}$ denotes the composition inverse and

$$\mathcal{G}_{\mu_A}(z) := \int_I dx \, \frac{1}{z-x} \mu_A(x), \tag{2}$$

is the *Stieltjes transform*.

We point out that the correspondence in Table 1 is by no mean exhaustive: to cite a few, there is also a clear correspondence for the multiplicative convolution with the so-called $S$-transform [1], the combinatorial moments-cumulants relations [3], the entropy [4] between the classical and the free world.

Since the discovery of free probability, it was unclear if one could find other generalized notion of independence, until Speicher [5] proved that, *under specific assumptions*, there is only three possible notions for a non-commutative algebraic probability space: classical independence, freeness and boolean independence [6]. By relaxing the assumptions, it has been however possible to construct other type of convolutions see for example [7]. Despite Speicher's work, there has been several attend to construct a generalized convolution, with or without an underlying notion of independence, that would in particular interpolate between the classical convolution and the free convolution, namely to cite few important results: the *q-convolution* of Nica [8] (see also [9] for a similar but different $q$-convolution) which interpolates between the classical convolution at $q = 1$ and the free convolution at $q = 0$ but which seems to not preserved the positivity of the measures [10]; the *t-convolution* of Benaych-Georges and Lévy in [11], which interpolated between the classical convolution ($t = 0$) and the free convolution ($t \to \infty$) but for which it is not possible to construct a transform that linearizes the convolution and from which one can define cumulants at any order.

In this note, we construct another one-parameter convolution, called the *c-convolution* as a continuous interpolation between the classical convolution at $c = 0$ and the free convolution as $c \to \infty$. Our construction is similar to the one developed in [8] in the sense that we construct an operator that interpolates naturally between the moment generating function and the exponential of (the integral of) the *R-transform*. Our $c$-convolution is technically defined on a set larger than the set of probability distribution and it is still an open question to know if it preserves positivity, nevertheless we show that several objects (see [12] [13] and [14] [15] [16] [17]) that have appeared before in the RMT literature at a specific limit $\frac{N\beta}{2} \to c$, called the *high temperature* regime, admit a simple interpretation in terms of our $c$-convolution.

Schematically our construction is as follows (concepts and notations will be made more precise in the main text). We start with the rank one HCIZ integral at finite $N$ and fixed $\beta$ between a matrix with eigenvalues $\boldsymbol{a}$ and another with a single non-zero eigenvalue $t$:

$$\mathcal{I}_{\boldsymbol{a}}^{(\beta)}(t) = \mathbb{E}_{\boldsymbol{v}} \left[ e^{(\boldsymbol{v}^* \underline{\boldsymbol{a}} \boldsymbol{v})t} \right] = \mathbb{E}_X \left[ e^{Xt} \right], \tag{3}$$

where $\boldsymbol{v}$ is the generalization to all $\beta > 0$ of a unit vector with real , complex, or quaternionic entries averaged over the corresponding sphere. We have introduced the random variable $X := \boldsymbol{v}^* \underline{\boldsymbol{a}} \boldsymbol{v}$ that

we will call the discrete Markov-Krein transform of $\boldsymbol{a}$. The rank one HCIZ integral is then the moment generating function of this variable. As $N \to \infty$ with fixed $\beta$ the variable $X$ concentrates on its average value and the Markov-Krein transformation is not very useful, but, as we will see, the variable $X$ converges to a non trivial measure as $N \to \infty$ with fixed $c := N\beta/2$. Our $c$-convolution will then be the (classical) convolution of Markov-Krein transforms, it naturally interpolates between the classical convolution ($c \to 0$) and the free convolution ($c \to \infty$).

In Section 2, we review several results concerning this HCIZ integral in the classical regime ($\beta > 0$), that will be useful to have a better understanding of our $c$-convolution, we focus on the rank one HCIZ as it is our main object of study. Section 3 is technically independant of our construction of the $c$-convolution and can be read independently, we show that we can make sense of the HCIZ for negative value of the parameter $\beta$, in particular we show that $\beta = -2$ is linked to the *finite free convolution* of Marcus [18] [19], many of the properties of the finite free convolution will have a clear analogous in the high temperature regime. In Section 4, we define the HCIZ in the high temperature regime and derived its properties, a particular focus is put on the Markov-Krein transform. Eventually, in Section 5, we introduce and discuss the properties of the $c$-convolution and derived several examples of $c$-convoluted objects.

**Acknowledgements:** We are very grateful to Jean-Philippe Bouchaud and Satya Majumdar for preliminary discussions and useful comments.

## 2 Review of some results concerning the rank one Harish-Chandra-Itzykson-Zuber Integral

### 2.1 Few words on the full rank case

In the 80', Itzykson and Zuber re-discovered Harish-Chandra's work on integrals over lie groups [20], in the context of random matrix theory (RMT). Such integrals are now referred as Harish-Chandra -Itzykson-Zuber (HCIZ in short) integrals, also known in the literature as angular/spherical integrals and as multivariate Bessel function. If we denote by $\beta = 1, 2, 4$ and $\mathbf{A}$ and $\mathbf{B}$ two $N \times N$ self-adjoint matrices with real, complex, quaternionic entries respectively, the HCIZ reads[1]:

$$\mathcal{I}^{(\beta)}(\mathbf{A}, \mathbf{B}) := \int_{\mathbf{G} \in \mathsf{G}^{(\beta)}} \mathcal{D}\mathbf{G}\, e^{\operatorname{Tr} \mathbf{A}\mathbf{G}\mathbf{B}\mathbf{G}^*} , \qquad (4)$$

where $\mathsf{G}^{(\beta)} = \mathsf{O}, \mathsf{U}, \mathsf{Sp}$ are respectively the orthogonal/unitary/symplectic $N$-dimensional groups.

From the spectral decomposition of $\mathbf{A}$ and $\mathbf{B}$, it is clear that the HCIZ integral only depends on their eigenvalues $\boldsymbol{a}$ and $\boldsymbol{b}$, so that we will denote it by $\mathcal{I}^{(\beta)}(\boldsymbol{a}, \boldsymbol{b})$ in the following. One may note also that since the vector of eigenvalues is unique up to permutation, the HCIZ integral is necessary a symmetric function in each argument $\boldsymbol{a}$ and $\boldsymbol{b}$.

In particular in the unitary case ($\beta = 2$), Itzykson and Zuber [21] have shown the famous formula bearing their name:

$$\mathcal{I}^{(2)}(\boldsymbol{a}, \boldsymbol{b}) = \left( \prod_{i=1}^{N-1} i! \right) \frac{\det(e^{a_i b_j})}{\Delta(\boldsymbol{a})\Delta(\boldsymbol{b})} , \qquad (5)$$

where $\Delta(\boldsymbol{a}) := \prod_{i<j}(a_i - a_j)$, is the Vandermonde determinant. The HCIZ integrals has applications in problems directly linked to random matrix theory (RMT) such as the study of the sum of invariant

---

[1]Note that some authors define the HCIZ integral with a constant $c_{N,\beta}$ in the exponential function that can be absorbed in one of the matrix $\mathbf{A}$ or $\mathbf{B}$.

ensembles [22] [23] [24], the development of large deviation principles [25], the study of the so-called orbital beta process [26], and also is linked to the enumeration of Hurwitz number in algebraic geometry [27] [28] and to quantum ergodic transport ([29]), to cite few recent results.

It is then tempting to try to generalize this formula for arbitrary positive $\beta$, just like one can study the eigenvalues distribution of $\beta$ ensembles in RMT for general $\beta$ [1]. There are several possible natural choices to define the HCIZ "integral"[2] for a generic $\beta > 0$ which lead all to the same result: a natural candidate is to see it as the symmetric eigenfunction of the so called Calogero-Moser operator normalized to unity whenever $\boldsymbol{a}$ or $\boldsymbol{b} = (0, \ldots, 0)$. One can then show [30] that the HCIZ integral admits the following representation for general $\beta > 0$:

$$\mathcal{I}^{(\beta)}(\boldsymbol{a}, \boldsymbol{b}) = \sum_{k=0}^{\infty} \sum_{|\lambda|=k} d_\lambda \, \mathrm{j}_\lambda^{\left(\frac{2}{\beta}\right)}(\boldsymbol{a}) \, \mathrm{j}_\lambda^{\left(\frac{2}{\beta}\right)}(\boldsymbol{b}) \,, \tag{6}$$

where the second sum is made over all *partitions of size $k$*: that is $\lambda = (\lambda_1, \lambda_2, \ldots)$ is a sequence of non-increasing integer such that $\sum \lambda_i = k$, $d_\lambda := \prod_{j=1}^N \frac{\Gamma\left(\frac{\beta}{2}(N-j+1)\right)}{\Gamma\left(\frac{\beta}{2}(N-j+1)+\lambda_j\right)}$ and the $\mathrm{j}_\lambda^{\left(\frac{2}{\beta}\right)}(\boldsymbol{a})$ are the so called "P" *Jack polynomials* index by the partition $\lambda$. The Jack polynomials are a one-parameter generalization of the *Schur Polynomials*, which corresponds to the case $\beta = 2$. At $\beta = 1$ (resp. $\beta = 4$), the Jack polynomials are the real (resp. quaternionic) *zonal polynomials*. We refer to [31] and [32] for properties concerning these polynomials.

## 2.2 The rank-one case

In this section and in the rest of the article, we fix one matrix to be of rank one, that is we have $\boldsymbol{b} = (t, 0, \ldots, 0)$, and we denote by:

$$\mathcal{I}_{\boldsymbol{a}}^{(\beta)}(t) := \mathcal{I}^{(\beta)}\left(\boldsymbol{a}, (t, 0, \ldots, 0)\right) \,, \tag{7}$$

the corresponding HCIZ integral that we see as a function of $t$ given the vector $\boldsymbol{a}$. The main reason to study this regime is that the large $N$ behavior of the rank one HCIZ integral is very different from the full rank case, which is known to satisfy a complex variational principle [33] [34], where analytical results are hard to obtain, except in some specific cases [35]. Specifying to the rank one case will greatly simplify results obtained for the full rank case and as a consequence we review known and lesser known formulas in the literature for the rank one HCIZ; namely the power sum representation (9), the operator differential representation (21), the inverse Laplace representation (25), the spherical Dirichlet average representation (30) and the moment generating function representation (35).

### 2.2.1 Power sum representation

We have from [32] the following simplification for the Jack polynomial:

$$\mathrm{j}_\lambda^{\left(\frac{t}{\beta}\right)}(t, 0, \ldots, 0) = \delta_{\lambda,k} \left(\frac{\beta}{2}\right)^k \frac{t^k}{k!} \,, \tag{8}$$

where $\delta_{\lambda,k} = 1$ if $\lambda = (k, 0, \ldots)$ and 0 otherwise. This greatly simplifies the expansion (6) and we have:

---

[2]by considering other values for $\beta$, we lack an Haar integral representation, but we will still call our object of interest the HCIZ "integral"

$$\boxed{\mathcal{I}_{\boldsymbol{a}}^{(\beta)}(t) = \sum_{k=0}^{\infty} \frac{\Gamma(\frac{N\beta}{2})}{\Gamma(\frac{N\beta}{2}+k)} \mathrm{g}_k^{(\frac{2}{\beta})}(\boldsymbol{a})t^k}, \tag{9}$$

where $\mathrm{g}_k^{(\frac{2}{\beta})}(\boldsymbol{a}) := \frac{1}{k!}\left(\frac{\beta}{2}\right)^k \mathrm{j}_k^{(\frac{2}{\beta})}(\boldsymbol{a})$. These normalized Jack polynomials admit a simple formula for their generating function, which can be taken as their definition:

$$\prod_{i=1}^{N}(1-a_i t)^{-\frac{\beta}{2}} = \sum_{k=0}^{\infty} \mathrm{g}_k^{(\frac{2}{\beta})}(\boldsymbol{a})t^k. \tag{10}$$

In particular, we see that if for $m \in \mathbb{N}$, we denote by $\boldsymbol{a}^{\otimes m} = (a_1, \ldots, a_1, \ldots, a_N, \ldots, a_N)$ the vector of size $mN$ obtained by making $m$ copies of the entries of the vector $\boldsymbol{a}$, we have:

$$\mathrm{g}_k^{(\frac{2}{\beta})}(\boldsymbol{a}) = \mathrm{g}_k^{(\frac{2m}{\beta})}(\boldsymbol{a}^{\otimes m}),$$

from which we derive the following $\beta \leftrightarrow N$ symmetry satisfied by the rank one HCIZ:

$$\mathcal{I}_{\boldsymbol{a}}^{(\beta)}(t) = \mathcal{I}_{\boldsymbol{a}^{\otimes m}}^{(\frac{\beta}{m})}(t). \tag{11}$$

In particular if $\beta$ is an integer we can always reduce to the $\beta = 1$ case since we have:

$$\mathcal{I}_{\boldsymbol{a}}^{(\beta)}(t) = \mathcal{I}_{\boldsymbol{a}^{\otimes\beta}}^{(1)}(t) \qquad \text{(for } \beta \text{ integer)}. \tag{12}$$

We state here another property of the normalized Jack polynomial $\mathrm{g}_k^{(\frac{2}{\beta})}(.)$ that will be useful later on: the *power sum symmetric polynomials* are defined for an integer $k$ by:

$$\mathrm{p}_k(\boldsymbol{a}) := \sum_{i=1}^{N} a_i^k, \tag{13}$$

that is they are the unnormalized moments of the discrete measure $\mu_{\boldsymbol{a}}(x) = \frac{1}{N}\sum_i \delta(x - a_i)$, where $\delta(.)$ is the Dirac mass distribution. They admit the following simple formula for their generating function which follows from the power sum expansion of the logarithm:

$$\log\left(\prod_{i}^{N}(1-a_i t)^{-1}\right) = \sum_{k=1}^{\infty} \frac{t^k}{k}\mathrm{p}_k(\boldsymbol{a}). \tag{14}$$

Combining (10) and (14), we can decompose the $\mathrm{g}_k^{(\frac{2}{\beta})}(.)$ in terms of the power sum polynomials which gives:

$$\mathrm{g}_k^{(\frac{2}{\beta})}(\boldsymbol{a}) = \sum_{1j_1+\cdots+kj_k=k}\left(\frac{\beta}{2}\right)^{j_1+\cdots+j_k}\prod_{i=1}^{k}\frac{\mathrm{p}_i(\boldsymbol{a})^{j_i}}{i^{j_i}j_i!}, \tag{15}$$

the first terms are given by:

1. $\mathrm{g}_0^{(\frac{2}{\beta})}(\boldsymbol{a}) = 1$

2. $\mathrm{g}_1^{(\frac{2}{\beta})}(\boldsymbol{a}) = \frac{\beta}{2}\mathrm{p}_1(\boldsymbol{a})$

3. $g_2^{(\frac{2}{\beta})}(\boldsymbol{a}) = \frac{1}{2}\left(\frac{\beta}{2}p_2(\boldsymbol{a}) + \left(\frac{\beta}{2}p_1(\boldsymbol{a})\right)^2\right)$

4. $g_3^{(\frac{2}{\beta})}(\boldsymbol{a}) = \frac{1}{3}\frac{\beta}{2}p_3(\boldsymbol{a}) + \frac{1}{2}\left(\frac{\beta}{2}p_2(\boldsymbol{a})\right)\left(\frac{\beta}{2}p_1(\boldsymbol{a})\right) + \frac{1}{6}\left(\frac{\beta}{2}p_1(\boldsymbol{a})\right)^3$

and we have the recurrence relation given by:

$$k\, g_k^{(\frac{2}{\beta})}(\boldsymbol{a}) = \frac{\beta}{2}\sum_{l=1}^{k} g_{k-l}^{(\frac{2}{\beta})}(\boldsymbol{a})\, p_l(\boldsymbol{a})\,. \tag{16}$$

**Remark:** Note that in the unitary case, this simplifies to:

$$\mathcal{I}_{\boldsymbol{a}}^{(2)}(t) = \sum_{k=0}^{\infty}\left(\frac{(N-1)!}{(k+N-1)!}h_k(\boldsymbol{a})\right)t^k\,, \tag{17}$$

where the $h_k(\boldsymbol{a})$ are the *complete homogeneous symmetric polynomials*:

$$h_k(\boldsymbol{a}) := \sum_{1\le j_1\le\cdots\le j_k\le N} a_{j_1}\ldots a_{j_k}\,. \tag{18}$$

This power sum expression in this unitary case can actually be derived from the Itzykson Zuber formula (5) using the Brézin-Hikami trick [36] and the identity: $h_k(\boldsymbol{a}) = \left(\sum_{i=1}^{N}\prod_{j\neq i}\frac{a_i^{k+N-1}}{a_j-a_i}\right)$, whenever all the $a_i$ are distinct.

### 2.2.2 Differential operator representation

Note that the coefficient $\frac{\Gamma(\frac{N\beta}{2})}{\Gamma(\frac{N\beta}{2}+k)}$ is precisely the inverse of the coefficient of

$$(-1)^k\frac{d^k}{dt^k}\left[t^{-\frac{N\beta}{2}}\right] = \frac{\Gamma(\frac{N\beta}{2}+k)}{\Gamma(\frac{N\beta}{2})}t^{-\frac{N\beta}{2}-k}\,. \tag{19}$$

By factorizing by $t^{-\frac{N\beta}{2}}$ and using the formula for the generating function of the Jack polynomials (10), we get the following differential operator form for the HCIZ rank one integral, relating the characteristic polynomial raised to the power $-\frac{\beta}{2}$ of the matrix with eigenvalues $\boldsymbol{a}$ denoted by:

$$U_{\boldsymbol{a}}^{(\beta)}(z) := \prod_{i=1}^{N}(z-a_i)^{-\frac{\beta}{2}} = e^{-\frac{N\beta}{2}\int du\,\log(z-u)\mu_{\boldsymbol{a}}(u)}\,, \tag{20}$$

with $\mu_{\boldsymbol{a}}(x) = \frac{1}{N}\sum_{i=1}^{N}\delta(x-a_i)$; with the one of the null matrix:

$$\boxed{U_{\boldsymbol{a}}^{(\beta)}(z) = \mathcal{I}_{\boldsymbol{a}}^{(\beta)}(-D)\,z^{-\frac{N\beta}{2}}}\,, \tag{21}$$

where $D^k := d^k/dz^k$. Equation (21) could have been taken as an alternative definition for the rank one HCIZ integral for general $\beta > 0$.

### 2.2.3 Inverse Laplace representation

From the power sum relation (9), we can express the HCIZ integral in terms of the inverse Laplace transform $\mathcal{L}_p^{-1}[.]$, using (10), we have for $t > 0$:

$$\mathcal{I}_{\boldsymbol{a}}^{(\beta)}(t) = \left(\frac{\Gamma(\frac{N\beta}{2})}{t^{\frac{N\beta}{2}-1}}\right) \sum_{k=0}^{\infty} g_k^{(\frac{2}{\beta})}(\boldsymbol{a}) \frac{1}{\Gamma(\frac{N\beta}{2}+k)} t^{\frac{N\beta}{2}+k-1}, \tag{22}$$

$$\mathcal{I}_{\boldsymbol{a}}^{(\beta)}(t) = \left(\frac{\Gamma(\frac{N\beta}{2})}{t^{\frac{N\beta}{2}-1}}\right) \sum_{k=0}^{\infty} g_k^{(\frac{2}{\beta})}(\boldsymbol{a}) \, \mathcal{L}_z^{-1}\left[\frac{1}{z^{\frac{N\beta}{2}+k}}\right](t), \tag{23}$$

$$\mathcal{I}_{\boldsymbol{a}}^{(\beta)}(t) = \left(\frac{\Gamma(\frac{N\beta}{2})}{t^{\frac{N\beta}{2}-1}}\right) \mathcal{L}_z^{-1}\left[\frac{1}{z^{\frac{N\beta}{2}}} \sum_{k=0}^{\infty} g_k^{(\frac{2}{\beta})}(\boldsymbol{a}) \frac{1}{z^k}\right](t), \tag{24}$$

and therefore by applying the generating function formula for $\frac{1}{z}$, we get the following Inverse Laplace representation:

$$\boxed{\mathcal{I}_{\boldsymbol{a}}^{(\beta)}(t) = \left(\frac{\Gamma(\frac{\beta}{2}N)}{t^{\frac{\beta}{2}N-1}}\right) \mathcal{L}_z^{-1}\left[U_{\boldsymbol{a}}^{(\beta)}(z)\right](t)} \qquad (t > 0), \tag{25}$$

with $U_{\boldsymbol{a}}^{(\beta)}(.)$ defined in (20), which gives in explicit form:

$$\mathcal{I}_{\boldsymbol{a}}^{(\beta)}(t) = \left(\frac{\Gamma(\frac{\beta}{2}N)}{t^{\frac{\beta}{2}N-1}}\right) \frac{1}{2\pi i} \int_{\gamma-i\infty}^{\gamma+i\infty} dz \, e^{tz} \prod_{i=1}^{N} (z-a_i)^{-\frac{\beta}{2}} \qquad (t > 0 \text{ and } \gamma > a_{\max}). \tag{26}$$

**Remark:** For the orthogonal and unitary cases, this formula could have been deduced from the definition of the HCIZ integral, by use of the Gaussian integration, see for example [1] and for the case $\beta > 0$ this can be deduce from the spiked $\beta$-Wishart ensemble of [37].

**Remark:** Note that from this expression, we clearly see the $\beta \leftrightarrow N$ symmetry (12).

### 2.2.4 Spherical Dirichlet integral representation

Another path to generalize the rank one HCIZ integral to arbitrary $\beta$ is to express it as an average of a simple function with respect to some $\beta$ dependent measure. In the classical case $\beta = 1, 2, 4$, from the definition (4), when the matrix $\mathbf{B}$ is a projector of rank one, we can re-express the HCIZ integral as:

$$\mathcal{I}_{\boldsymbol{a}}^{(\beta)}(t) = \int_{\mathbb{S}_\beta^{N-1}} d\boldsymbol{\sigma} \, e^{t \sum_{i=1}^{N} a_i \left(\sum_{b=1}^{\beta} \sigma_{i,b}^2\right)} \qquad (\beta = 1, 2, 4), \tag{27}$$

with $\mathbb{S}_\beta^{N-1} := \left\{\boldsymbol{\sigma} \in \mathbb{R}^{N\beta} \mid \sum_{i=1}^{N} \sum_{b=1}^{\beta} \sigma_{i,\beta}^2 = 1\right\}$, in particular $\mathbb{S}_1^{N-1} = \mathbb{S}^{N-1}$ is the usual $N$-dimensional real sphere. We can then make $N$ times the $\beta$ polar change of coordinates $x_i^2 = \sum_{b=1}^{\beta} \sigma_{i,\beta}^2$, from which we find:

$$\mathcal{I}_{\boldsymbol{a}}^{(\beta)}(t) \propto \int_{\mathbb{S}^{N-1}} d\boldsymbol{x} \, |x_1 \dots x_N|^{\beta-1} e^{t \sum_{i=1}^{N} a_i x_i^2} \qquad (\beta = 1, 2, 4). \tag{28}$$

Following [38], we can generalize the above equation to arbitrary $\beta > 0$ by introducing the following $\alpha$ *spherical Dirichlet distribution* with $\alpha \geq 0$ defined on the real sphere $\mathbb{S}^{N-1}$ as

$$\mu^{(\alpha)}(\boldsymbol{x}) := \frac{\Gamma(\frac{N\alpha}{2})}{\Gamma(\frac{\alpha}{2})^N} |x_1 \ldots x_N|^{\alpha-1} . \tag{29}$$

Using this measure, we could define the rank one HCIZ integral for arbitrary $\beta > 0$ by

$$\boxed{\mathcal{I}_{\boldsymbol{a}}^{(\beta)}(t) = \mathbb{E}_{\boldsymbol{v} \sim \mu^{(\beta)}} \left[ e^{(\boldsymbol{v}^* \underline{\boldsymbol{a}} \boldsymbol{v})t} \right]} . \tag{30}$$

As explained nicely in [38], the parameter $\alpha$ determines how the mass is concentrated on the sphere and we have in particular:

1. $\mu^{(1)}(.)$ is the uniform measure on the sphere

2. $\mu^{(0)}(\boldsymbol{x}) = \frac{1}{2N} \sum_{i=1}^{2N} \delta(\boldsymbol{x} \pm \boldsymbol{e_i})$, where $\boldsymbol{e_i}$ is the $i^{th}$ vector of the canonical basis.

3. $\mu^{(\infty)}(\boldsymbol{x}) = \frac{1}{2^N} \sum_{i=1}^{2^N} \delta\left(\boldsymbol{x} - \frac{1}{\sqrt{N}}(\pm 1, \ldots, \pm 1)\right)$ .

This intuitive generalization only works for the rank one case and is therefore less general than our definition (6) using Jack polynomials. It is important to verify that (30) can derived from our original definition. As noted by the above authors [38], if we denote by $\underline{\boldsymbol{a}} = \mathrm{Diag}(\boldsymbol{a}) = \mathrm{Diag}(a_1, \ldots, a_N)$, we have:

$$U_{\boldsymbol{a}}^{(\beta)}(z) = \int_{\mathbb{S}^{N-1}} d\boldsymbol{x} \, (z - \boldsymbol{v}^* \underline{\boldsymbol{a}} \boldsymbol{v})^{-\frac{N\beta}{2}} \mu^{(\beta)}(\boldsymbol{x}) , \tag{31}$$

so with the inverse Laplace representation (25), we get:

$$\mathcal{I}_{\boldsymbol{a}}(t) = \left( \frac{\Gamma(\frac{\beta}{2}N)}{t^{\frac{\beta}{2}N-1}} \right) \mathcal{L}_z^{-1} \left[ (z)^{-N\frac{\beta}{2}} \right] \mathbb{E}_{\boldsymbol{v} \sim \mu^{(\beta)}} \left[ e^{(\boldsymbol{v}^* \underline{\boldsymbol{a}} \boldsymbol{v})t} \right] , \tag{32}$$

from which we recover (30).

### 2.2.5 Moment generating function representation

We finish this section with an important formula for the rest of this article: if we now make the following change of variable $\boldsymbol{d} = (d_i, \ldots, d_N)$ with $d_i = v_i^2$ in (30), then we have:

$$\mathcal{I}_{\boldsymbol{a}}^{(\beta)}(t) = \mathbb{E}_{\boldsymbol{d} \sim \mu_{\mathrm{Dir}}} \left[ e^{t\boldsymbol{a}^* \boldsymbol{d}} \right] , \tag{33}$$

where $\boldsymbol{d}$ follows the (planar) Dirichlet distribution with parameter $(\frac{\beta}{2}, \ldots, \frac{\beta}{2})$: its probability density function is defined over the simplex $\Delta = \{x_i \in (0,1) | \sum x_i = 1\}$ and given by:

$$\mu_{\mathrm{Dir}}(\boldsymbol{x}) = \frac{1}{C_{\beta,N}} |x_1 \ldots x_N|^{\frac{\beta}{2}-1} , \tag{34}$$

then doing the change of variable $X = \boldsymbol{a}^* \boldsymbol{d}$ allows us to represent the HCIZ integral as a moment generating function:

$$\boxed{\mathcal{I}_{\boldsymbol{a}}^{(\beta)}(t) = \mathbb{E}_{X \sim \mathcal{M}_{\frac{N\beta}{2}, \boldsymbol{a}}} \left[ e^{tX} \right]} . \tag{35}$$

The distribution $\mathcal{M}_{\frac{N\beta}{2},\boldsymbol{a}}$ is known as a *mean Dirichlet process* or as the *(discrete) Markov-Krein transform* (MKT) of the vector $\boldsymbol{a}$, with parameter $\frac{N\beta}{2}$. One should think of the transformed variable $X$ as a random convex combination of the $a_i$'s, its support is naturally given by the extreme values of $\boldsymbol{a}$ namely $[a_{\min}, a_{\max}]$. By the symmetry of the Dirichlet process the first moment is preserved: $\mathbb{E}[X] = \frac{1}{N}\sum_{i=1}^{N} a_i$. and for $t > 0$, we see that the bounds:

$$e^{t\,a_{\min}} \leq \mathcal{I}_{\boldsymbol{a}}^{(\beta)}(t) \leq e^{t\,a_{\max}} \qquad (t > 0), \qquad (36)$$

which is immediate for $\beta = 1, 2, 4$ from the definition of the HCIZ integral, is preserved.

Next we give a formula that we will prove latter in a more general context, relating the vector $\boldsymbol{a}$ to the distribution $\mathcal{M}_{\frac{N\beta}{2},\boldsymbol{a}}$:

$$\int_{a_{\min}}^{a_{\max}} dx\,(z - x)^{-\frac{N\beta}{2}} \mathcal{M}_{\frac{N\beta}{2},\boldsymbol{a}}(x) = U_{\boldsymbol{a}}^{(\beta)}(z). \qquad (37)$$

## 2.3 Large $N$ behavior of the rank one HCIZ

### 2.3.1 $\beta > 0$ and relation with free probability

As explained in the introduction of this section, the main reason to study the rank one HCIZ integral is its large $N$ behavior. In particular, it is known for the three classical value $\beta = 1, 2, 4$ that if we denote by $\boldsymbol{\gamma} = \mathrm{Eigen}(\underline{\boldsymbol{a}} + \mathbf{G}'\underline{\boldsymbol{b}}\mathbf{G}'^*)$, with $\underline{\boldsymbol{a}} = \mathrm{Diag}(\boldsymbol{a}) = \mathrm{Diag}(a_1, \ldots, a_N)$ and similarly for $\underline{\boldsymbol{b}}$, we have from the property of the Haar measure, the following formula for the rank one HCIZ:

$$\mathbb{E}_{\mathbf{G}' \in \mathsf{G}^{(\beta)}}\left[\mathcal{I}_{\boldsymbol{\gamma}}^{(\beta)}(t)\right] = \mathcal{I}_{\boldsymbol{a}}^{(\beta)}(t)\,\mathcal{I}_{\boldsymbol{b}}^{(\beta)}(t) \qquad (\beta = 1, 2, 4). \qquad (38)$$

In the large $N$ limit, we assume that the spectral measure $\mu_{\boldsymbol{a}}(x) := \frac{1}{N}\sum_{i=1}^{N}\delta(x - a_i)$ converges[3] to a compactly supported deterministic measure $\mu_A$ such that $\min \boldsymbol{a} \to a_{\min}$ and $\max \boldsymbol{a} \to a_{\max}$, where $a_{\min}$ and $a_{\max}$ are the left and right extremities of the support of the measure $\mu_A$.

We expect to have some self-averaging in the LHS of (38), so that we can remove the expectation, making the logarithm of the HCIZ additive for the free convolution and therefore directly connected to the famous $\mathcal{R}$-transform of RMT. To establish such relation, we perform a standard saddle point analysis in (26) for $\beta > 0$:

$$\mathcal{I}_{\boldsymbol{a}}^{(\beta)}(Nt) = \left(\frac{\Gamma(\frac{\beta}{2}N)}{(Nt)^{\frac{\beta}{2}N-1}}\right) \frac{1}{2\pi i}\int_{\gamma-i\infty}^{\gamma+i\infty} dz\, e^{Ntz - \frac{\beta}{2}\sum_{i=1}^{N}\log(z-a_i)}, \qquad (39)$$

$$\mathcal{I}_{\boldsymbol{a}}^{(\beta)}(Nz) = \frac{1}{2\pi i}\int_{\gamma-i\infty}^{\gamma+i\infty} dz\, e^{N\mathcal{H}^{(\beta)}(z,t)}, \qquad (40)$$

with:

$$\begin{aligned}
\mathcal{H}^{(\beta)}(z,t) = {} & tz - \frac{\beta}{2}\int dx\,\log(z-x)\,\mu_{\boldsymbol{a}}(x) - \frac{\beta}{2}\log(t) \\
& + \frac{1}{N}\left(\log\left(\Gamma\left(\frac{\beta}{2}N\right)\right) - \left(\frac{\beta}{2}N - 1\right)\log(N)\right) + \frac{1}{N}\log(t),
\end{aligned} \qquad (41)$$

Now we have by Stirling formula that:

$$\lim_{N\to\infty} \frac{1}{N}\left(\log\left(\Gamma\left(\frac{\beta}{2}N\right)\right) - \left(\frac{\beta}{2}N - 1\right)\log N\right) + \frac{1}{N}\log z = \frac{\beta}{2}\log\frac{\beta}{2} - \frac{\beta}{2}, \qquad (42)$$

---

[3] for simplicity we write $\boldsymbol{a}$ instead of $\boldsymbol{a}_{(N)}$ even though the vector $\boldsymbol{a}$ is $N$-dependent

and since we have $\mu_{\boldsymbol{a}} \to \mu_A$ we have:

$$\lim_{N\to\infty} \frac{1}{N} \log \mathcal{I}_{\boldsymbol{a}}^{(\beta)}(Nt) = \mathcal{H}^{(\beta)}(z^*(t), t) , \tag{43}$$

with

$$\mathcal{H}^{(\beta)}(z, t) \simeq zp - \frac{\beta}{2} \int dx \, \log(z - x) \, \mu_{\boldsymbol{a}}(x) - \frac{\beta}{2} \log(t) + \frac{\beta}{2} \log \frac{\beta}{2} - \frac{\beta}{2} ,$$

and $p^*(z)$ solution of :

$$\begin{cases} \partial_z \mathcal{H}^{(\beta)}(z, t) & = 0 \\ t - \frac{\beta}{2} \mathcal{G}_{\mu_A}(z) & = 0 \end{cases} , \tag{44}$$

with $\mathcal{G}_\mu(.)$ is defined in (2). That is $z^*(t) = \mathcal{G}_{\mu_A}^{(-1)} \left( \frac{2}{\beta} t \right)$ for $t$ close enough to the origin. One may notice then:

$$\frac{d}{dz} \mathcal{H}^{(\beta)}(z^*(t), t) = \mathcal{G}_{\mu_A}^{(-1)} \left( \frac{2}{\beta} t \right) + \frac{2}{\beta} t \frac{d}{dt} \mathcal{G}_{\mu_A}^{(-1)} \left( \frac{2}{\beta} t \right) - (\frac{2}{\beta} t) \frac{d}{dt} \mathcal{G}_{\mu_A}^{(-1)} \left( \frac{2}{\beta} t \right) - \frac{\beta}{2} \frac{1}{t} , \tag{45}$$

$$\frac{d}{dt} \mathcal{H}^{(\beta)}(z^*(t), t) = \mathcal{G}_{\mu_A}^{(-1)} \left( \frac{2}{\beta} t \right) - \frac{\beta}{2} \frac{1}{t} , \tag{46}$$

so that at the end we have the following simple formula:

$$\lim_{N\to\infty} \frac{1}{N} \frac{d}{dt} \log \mathcal{I}_{\boldsymbol{a}}^{(\beta)}(Nt) = \mathcal{R}_{\mu_A} \left( \frac{2}{\beta} t \right) \qquad (t \text{ close to } 0) , \tag{47}$$

where $\mathcal{R}_{\mu_A}(.)$ is defined in (1). This result was first derived for $\beta = 1, 2$ by Parisi [39] and made rigorous by Guionnet and Maïda [40] for $\beta = 1, 2$ using Gaussian concentration and under a more general setting. In particular, the asymptotic for all $t$ and not just close to origin is derived and one can see that there is a phase transition at a certain $t^*$ above which the asymptotic (47) is no more true, we refer to [40] for more details.

**Remark:** Note that for $\beta$ integer, this is consistent with the $\beta \leftrightarrow N$ symmetry (12) since $\mu_{\boldsymbol{a}^{\otimes\beta}} \to \mu_A$.

### 2.3.2  Infinite temperature regime ($\beta \to 0$) and classical convolution

In the previous subsection, the parameter $\beta$ was fixed to real positive values. The aim of this section is to describe the extreme value zero. To get the behavior for this value, we will use equivalently the limiting behavior of the Jack polynomials and the Dirichlet average representation (30).

By the recurrence relation (16) satisfied by the $g_k(.)$ and using properties of the gamma function, we have for $k > 0$:

- $\lim_{\beta\to 0} \left( \frac{2}{\beta} \right) g_k^{(\frac{2}{\beta})}(\boldsymbol{a}) = \frac{\operatorname{Tr} \underline{\boldsymbol{a}}^k}{k}$ , where we recall $\underline{\boldsymbol{a}} := \operatorname{Diag}(\boldsymbol{a})$.

- $\lim_{\beta\to 0} \frac{\Gamma(\frac{\beta}{2}N)(\frac{\beta}{2})}{\Gamma(\frac{\beta}{2}N+k)} = \frac{1}{N(k-1)!}$ ,

so that in the end we get for the HCIZ integral:

$$\lim_{\beta \to 0} \mathcal{I}_{\boldsymbol{a}}^{(\beta)}(t) = \sum_{k=0}^{\infty} \frac{m_k(\boldsymbol{a})}{k!} t^k \,, \tag{48}$$

with $m_k(\boldsymbol{a}) := \frac{1}{N} \sum_{k=1}^{N} a_i^k$, is the $k^{th}$ moment of the (random) distribution $\mu_{\boldsymbol{a}}$.

This is also consistent will the Dirichlet average representation, since in this case the measure degenerates at the poles $\pm \boldsymbol{e}_i$ with $\boldsymbol{e}_i$ the $i^{th}$ canonical vector.

$$\frac{1}{2N} \sum \delta \left( \boldsymbol{v} \pm \boldsymbol{e}_i \right) e^{\sum_{i=1}^{N} a_i v_i^2 t} = \frac{1}{N} \sum_{i=1}^{N} e^{a_i t} \,, \tag{49}$$

$$\frac{1}{2N} \sum \delta \left( \boldsymbol{v} \pm \boldsymbol{e}_i \right) e^{\sum_{i=1}^{N} a_i v_i^2 t} = \sum_{k=0}^{\infty} \frac{m_k(\boldsymbol{a})}{k!} t^k \,, \tag{50}$$

that is we have:

$$\lim_{\beta \searrow 0} \mathcal{I}_{\boldsymbol{a}}^{(\beta)}(t) = \mathbb{E}_{X \sim \mu_A} \left[ e^{tX} \right] \,. \tag{51}$$

In other words in the $\beta$ goes to zero limit, the rank one HCIZ is nothing else than the classical generating function of the moments and under the same assumptions as in Section 2.3.1, this property is preserved by the limit $N \to \infty$. In the Markov-Krein language the variable $X$ can only take values $a_i$ each with probability $1/N$ hence its measure is equal to the discrete measure $\mu_{\boldsymbol{a}}$. In particular, its logarithm is the generating function of the *classical* cumulants, which is expected since in the theory of $\beta$-ensembles, the parameter $\beta$ measures the strength of the interactions between the eigenvalues, at $\beta = 0$ there is no interactions and one recovers classical objects. It is worth noting that if we denote by $\mathbf{P}$ a $N \times N$ permutation matrix, we can express the rank one HCIZ at $\beta = 0$ as an Haar integral:

$$\mathcal{I}_{\boldsymbol{a}}^{(0)}(t) = \int_{\mathbf{P} \in \mathsf{Sym}(N)} \mathcal{D}\mathbf{P} \, e^{t(\mathbf{P}\underline{\boldsymbol{a}}\mathbf{P}^*)_{11}} \,, \tag{52}$$

where $\mathcal{D}\mathbf{P}$ is the normalized (discrete) counting measure of the permutation group. This is actually a special case of the formula of the full-rank case, since we have:

$$\mathcal{I}^{(0)}(\boldsymbol{a}, \boldsymbol{b}) = \int_{\mathbf{P} \in \mathsf{Sym}(N)} \mathcal{D}\mathbf{P} \, e^{(\mathbf{P}\underline{\boldsymbol{a}}\mathbf{P}^*\underline{\boldsymbol{b}})} \,. \tag{53}$$

**Remark:** Similarly in the freezing regime ($\beta \to \infty$) we get:

$$\lim_{\beta \to \infty} \mathcal{I}_{\boldsymbol{a}}^{(\beta)}(t) = e^{\left( \frac{1}{N} \operatorname{Tr} \underline{\boldsymbol{a}} \right) t} \,. \tag{54}$$

In this limit and with this scaling, the HCIZ integral only captures the mean of the limiting distribution and as a consequence does not give much information on the complex structure of this regime where one expects the eigenvalues to "freeze" on a lattice, see for example [41].

# 3 Negative $\beta$ and finite free convolution

## 3.1 definition

It is tempting to generalize the HCIZ formula to negative value $\beta = -\gamma$, $\gamma > 0$. To do so, let's introduce the following generalization of the Jack polynomials:

$$\prod_{i=1}^{N}(1 - a_i t)^{\frac{\gamma}{2}} := \sum_{k=0}^{\infty} \mathrm{g}_k^{\left(-\frac{2}{\gamma}\right)}(\boldsymbol{a})t^k \,. \tag{55}$$

**Remark:** If $\gamma$ is even ($\gamma \in 2\mathbb{N}$), then we have $\mathrm{g}_k^{\left(-\frac{2}{\gamma}\right)}(\boldsymbol{a}) = 0$ for $k > \frac{N\gamma}{2}$, since the LHS is a polynomial in $t$.

In particular, we have for $\gamma = 2$:

$$\mathrm{g}_k^{(-1)}(\boldsymbol{a}) = \begin{cases} (-1)^k \mathrm{e}_k(\boldsymbol{a}) & \text{for } k \leq N \\ 0 & \text{otherwise,} \end{cases} \tag{56}$$

where the $\mathrm{e}_k(.)$ are the *elementary symmetric polynomials*:

$$\mathrm{e}_k(\boldsymbol{a}) := \sum_{1 \leq j_1 < \cdots < j_k \leq N} a_{j_1} \ldots a_{j_k} \,. \tag{57}$$

By Euler's formula:

$$\Gamma(1 - t) = \frac{\pi}{\Gamma(t)\sin \pi t} \qquad \text{for } t \in \mathbb{C}\backslash\, \mathbb{N}, \tag{58}$$

we can then formally define the rank one HCIZ integral for negative $\beta = -\gamma$ by simply taking (58) with the definition of the negative Jack polynomials (55) in (9). By singularity of the gamma function at negative integer, this extension of the definition of the HCIZ integral to negative value is, at $N$ fixed, only true for specific value of the parameter $\gamma$ due to the term:

$$\frac{\Gamma\left(\frac{N\gamma}{2} - k + 1\right)}{\Gamma\left(\frac{N\gamma}{2} + 1\right)} \mathrm{g}_k^{\left(-\frac{2}{\gamma}\right)}(\boldsymbol{a}) \,, \tag{59}$$

in the sum. For $\gamma$ even, thanks to the previous remark, we see that there is no problem since we can fix it to be equal to zero for $k > \frac{N\gamma}{2}$ and hence there is no singularity. So if we define by:

$$R_{(N)} := \left\{ \gamma \in \mathbb{R}_+ \text{ such that } \frac{N\gamma}{2} \notin \mathbb{N} \text{ or } \gamma \in 2\mathbb{N} \right\}, \tag{60}$$

the set of admissible value of $\gamma$, then we can define the HCIZ at negative value by:

$$\mathcal{I}_{\boldsymbol{a}}^{(-\gamma)}(-t) := \sum_{k=0}^{\infty} \frac{\Gamma\left(\frac{N\gamma}{2} - k + 1\right)}{\Gamma\left(\frac{N\gamma}{2} + 1\right)} \mathrm{g}_k^{\left(-\frac{2}{\gamma}\right)}(\boldsymbol{a})t^k \qquad \left(\text{for } \gamma \in R_{(N)}\right). \tag{61}$$

Similarly to the positive case, we have:

$$\mathrm{D}^k\, t^{\frac{N\gamma}{2}} = \frac{\Gamma\left(\frac{N\gamma}{2}+1\right)}{\Gamma\left(\frac{N\gamma}{2}-k+1\right)} t^{\frac{N\gamma}{2}-k}\,, \tag{62}$$

which leads us to the following operator differential representation:

$$\prod_{i=1}^{N}(z-a_i)^{\frac{\gamma}{2}} = \mathcal{I}_{\boldsymbol{a}}^{\left(-\frac{2}{\gamma}\right)}(-\mathrm{D})\, z^{\frac{N\gamma}{2}} \qquad \left(\text{for } \gamma \in R_{(N)}\right)\,. \tag{63}$$

Again for $\frac{N\gamma}{2} \in \mathbb{N}$, the RHS of (63) is a sum of derivatives of a polynomial and hence a polynomial whereas the LHS (for $\gamma \notin 2\mathbb{N}$) is a formal power sum and therefore strict equality is not possible. When $\gamma \in 2\mathbb{N}$, we have an equality between two polynomials.

**Remark:**  By the limits:

- $\lim_{\gamma\to 0} \frac{2}{\gamma}\, \mathrm{g}_k^{\left(-\frac{2}{\gamma}\right)}(\boldsymbol{a}) = -\frac{\mathrm{Tr}\,\boldsymbol{a}^k}{k}$

- $\lim_{\gamma\to 0} \Gamma\left(\frac{\gamma N}{2}\right)\frac{\gamma}{2} = \frac{1}{N(k-1)!}$

  we see that we have $\mathcal{I}_{\boldsymbol{a}}^{(0^-)}(-t) = \mathcal{I}_{\boldsymbol{a}}^{(0^+)}(t) = \sum_{k=0}^{\infty} \frac{m_k(\boldsymbol{a})}{k!} t^k$.

### 3.2  The special case $\gamma$ even

In the rest of this section, we look at the special case $\gamma \in 2\mathbb{N} = \{2, 4, 6, \dots\}$. It is immediate from the definition of the negative Jack polynomials that we have again a $\gamma \leftrightarrow N$ symmetry, in particular for $m \in \mathbb{N}$, $k < \frac{Nm\gamma}{2}$, using (56) we have:

$$\mathrm{g}_k^{\left(-\frac{2}{m\gamma}\right)}(\boldsymbol{a}) = (-1)^k \mathrm{e}_k\left(\boldsymbol{a}^{\otimes m}\right)\,, \tag{64}$$

so we can reduce to the case $\gamma = 2$ without any loss of generality. In this setting we have:

$$\mathcal{I}_{\boldsymbol{a}}^{(-2)}(-t) = \sum_{k=0}^{N} \frac{(N-k)!}{N!}(-1)^k \mathrm{e}_k(\boldsymbol{a}) t^k\,, \tag{65}$$

$$\mathcal{I}_{\boldsymbol{a}}^{(-2)}(-t) = \left(\frac{t^{N+1}}{N!}\right)\sum_{k=0}^{N}(N-k)!(-1)^k \mathrm{e}_k(\boldsymbol{a})\, t^{k-N-1}\,, \tag{66}$$

but since we have:

$$(N-k)!\, t^{k-N-1} = \mathcal{L}_z\left[z^{N-k}\right](t)\,, \tag{67}$$

where $\mathcal{L}_z\left[.\right]$ is the Laplace transform with respect to the variable $z$, we get:

$$\mathcal{I}_{\boldsymbol{a}}^{(-2)}(-t) = \left(\frac{t^{N+1}}{N!}\right) \mathcal{L}_z \left[\sum_{k=0}^{N} (-1)^k \mathrm{e}_k(\boldsymbol{a}) \, z^{N-k}\right] , \tag{68}$$

$$\mathcal{I}_{\boldsymbol{a}}^{(-2)}(-t) = \left(\frac{t^{N+1}}{N!}\right) \mathcal{L}_z \left[\prod_{i=1}^{N} (z - a_i)\right] , \tag{69}$$

$$\mathcal{I}_{\boldsymbol{a}}^{(-2)}(-t) = \left(\frac{t^{N+1}}{N!}\right) \int_0^\infty dz \, e^{-zt + \int du \, \log(z-u)\mu_{\boldsymbol{a}}(u)} . \tag{70}$$

This expression is the negative counterpart of (25). From (69) it is clear that in the large $N$ asymptotic the integral is dominated by the same saddle point as the one in Section 2.3.1, so under the same assumption as in Section 2.3.1, we directly conclude the following asymptotic:

$$\lim_{N\to\infty} -\frac{1}{N}\frac{d}{dt} \log \mathcal{I}_{\boldsymbol{a}}^{(-2)}(-Nt) = \mathcal{R}_{\mu_A}(t) . \tag{71}$$

The case for general $\gamma \in 2\mathbb{N}$ follow easily using the $\gamma \leftrightarrow N$ symmetry (64).

### 3.3 Link with finite free convolution

In [19] and [18] the authors have introduced the following convolution, known as the *finite free convolution*: Let $\mu_{\boldsymbol{a}}(x) = \frac{1}{N}\sum_{i=1}^{N} \delta(x - a_i)$ and $\mu_{\boldsymbol{b}}(x) = \frac{1}{N}\sum_{i=1}^{N} \delta(x - b_i)$ be two finite distributions of the same size $N$. Then, since we are at $\gamma = 2$, (63) simply becomes:

$$\prod_{i=1}^{N}(t - a_i) = \mathcal{I}_{\boldsymbol{a}}^{(-2)}(-\mathrm{D}) \, t^N , \tag{72}$$

and similarly for $\boldsymbol{b}$. Their *finite free convolution* denoted by:

$$\mu_{\boldsymbol{c}} = \mu_{\boldsymbol{a}} \boxplus_N \mu_{\boldsymbol{b}} , \tag{73}$$

is then defined as the unique, well behaved, finite $N$ probability measure on the (real) points $c_i$ which are solutions of:

$$\prod_{i=1}^{N}(t - c_i) = \mathcal{I}_{\boldsymbol{a}}^{(-2)}(-\mathrm{D}) \, \mathcal{I}_{\boldsymbol{b}}^{(-2)}(-\mathrm{D}) \, t^N . \tag{74}$$

We refer to [19] and [18] for several other formulations and properties of this convolution. In particular (74) can be restated as:

$$\mathcal{I}_{\boldsymbol{c}}^{(-2)}(t) = \mathcal{I}_{\boldsymbol{a}}^{(-2)}(t) \, \mathcal{I}_{\boldsymbol{b}}^{(-2)}(t) \qquad\qquad \mathrm{mod} \; t^{N+1} , \tag{75}$$

where mod $t^{N+1}$ means equality of the power series up to the $N^{th}$ term, which is obviously needed since we known that $\mathcal{I}_{\boldsymbol{c}}^{(-2)}(.)$ is a polynomial of order $N$ while the product in the right hand side (RHS) of (75) is a polynomial of order $2N$, where terms of order higher than $N$ do not contribute in (74). Now, under the same assumptions as in Section 2.3.1, taking the limit $N$ goes to infinity in (75), we can formally remove the mod $t^{N+1}$, so that together with the limit (71), we have:

$$\lim_{N \to \infty} (\mu_{\boldsymbol{a}} \boxplus_N \mu_{\boldsymbol{b}}) = \mu_A \boxplus \mu_B \,, \tag{76}$$

hence the name finite free convolution.

We conclude this section with another interesting point of view, detailed in [18], concerning the finite free convolution that will have a clear analogous in our construction of the $c$-convolution of Section 5. To each finite $N$ measure $\mu_{\boldsymbol{a}}$ we can associate a finite $N$ *complex* valued measure $\mu_{\boldsymbol{s}}(z) = \frac{1}{N} \sum_{i=1}^{N} \delta(z - s_i)$ that we call the *negative Markov-Krein transform*[4] of $\mu_{\boldsymbol{a}}$ such that we have:

$$\int_{\mathbb{C}} du \,(z - u)^N \mu_{\boldsymbol{s}}(u) = \prod_{i=1}^{N} (z - a_i) \,, \tag{77}$$

then plugging (77) in (69), one arrives at (see [18]):

$$\mathcal{I}_{\boldsymbol{a}}^{(-2)}(t) = \int_{\mathbb{C}} du \, e^{tu} \mu_{\boldsymbol{s}}(u) \qquad\qquad \mathrm{mod}\ t^{N+1} \,. \tag{78}$$

We note the clear correspondence between the $\beta > 0$ case and the $\beta = -2$, in particular we see that (77) is the negative counterpart of (37) at $\beta = -2$ while (78) is the negative counterpart of (35), we see that due to the lack of a Dirichlet representation, the negative Markov-Krein transform is complex valued. Nevertheless, (78) together with (75) indicates that the finite free convolution can be understood - up to a truncation operation - as a convolution of the negative Markov-Krein transforms.

# 4 HCIZ at the high temperature limit $\frac{N\beta}{2} \to c$

## 4.1 Definition and notations

From Section 2.3, we have seen that the HCIZ transform exhibits a drastic change of behavior in the parameter $\beta$ near the origin. As it is standard statistical physics (see for example [42] for a model linked to RMT), to introduce a continuous phase transition between the two regimes, we take $\beta$ going *slowly* to 0 by which we mean $\frac{N\beta}{2} \to c$, where $c \geq 0$ is a tunable parameter [5]. Since this limit only makes sense as $N$ goes to infinity, the goal of this subsection is to make precise what we mean by HCIZ at high temperature and show that most of the representations of Section 2.2 admit an high temperature counterpart.

Let's fix a *compactly* supported measure $\mu_A$ with support $I$. The corresponding $c$-HCIZ is defined by:

$$\mathcal{I}_{\mu}^{(c)}(t) := \sum_{k=0}^{\infty} \frac{\Gamma(c)}{\Gamma(c + k)} \mathrm{g}_k^{(c)}(\mu)\, t^k \,, \tag{79}$$

where the $\mathrm{g}_k^{(c)}(\mu)$ are defined by taking the $\frac{N\beta}{2} \to c$ in the power sum expansion of the normalized Jack polynomials (15) which gives:

---

[4]in [18], the distribution $\mu_{\boldsymbol{s}}$ is called the $U$-transform of the set $\boldsymbol{a}$

[5]Note that even though other scalings could have been chosen, this particular one has already been studied in the RMT literature in a completely different context [12] [14], and has shown to exhibit non-trivial limiting objects.

$$g_k^{(c)}(\mu) := \sum_{1j_1 + \cdots + kj_k = k} c^{j_1 + \cdots + j_k} \prod_{i=1}^{k} \frac{m_i(\mu)^{j_i}}{i^{j_i} j_i!} \,, \tag{80}$$

where $m_i(\mu)$ the $i^{th}$ moment of the measure $\mu$. They satisfy the recurrence:

$$k \, g_k^{(c)}(\mu) = c \sum_{l=1}^{k} g_{k-l}^{(c)}(\mu) \, m_l(\mu) \,. \tag{81}$$

We define the high temperature analog of $U_{\boldsymbol{a}}^{(\beta)}(z) := \det(z - \underline{\boldsymbol{a}})^{-\beta/2}$,

$$U_\mu^{(c)}(z) := \exp\left\{ -c \int_I dx \, \log(z - x)\mu(x) \right\} \,, \tag{82}$$

which by property of the logarithm, is analytical for all $\mathbb{C} \setminus (-\infty, a_{\max})$ [6] and can be equivalently represented as:

$$U_\mu^{(c)}(z) = \frac{1}{z^c} \sum_{k=0}^{\infty} g_k^{(c)}(\mu) \frac{1}{z^k} \,. \tag{83}$$

and is linked to Stieltjes transform by:

$$\mathcal{G}_\mu(z) = -\frac{1}{c} \frac{d}{dz} \log U_\mu^{(c)}(z) \,. \tag{84}$$

It is worth noting that from the usual Plemelj inversion formula (see (101) in Section 4.2), one may recover the original distribution thanks to the inversion formula:

$$\mu(x) = -\frac{1}{c\pi} \frac{d}{dx} \lim_{\eta \searrow 0} \mathfrak{Im} \log\{U_\mu^{(c)}(x - i\eta)\} \,, \tag{85}$$

where the derivative has to be understood in the distributional sense. Next, by doing the same derivation as in Section 2.2.3, we get that the following high temperature counterpart of (25):

$$\mathcal{I}_\mu^{(c)}(t) = \frac{\Gamma(c)}{t^{c-1}} \mathcal{L}_z^{-1}\left[ U_\mu^{(c)}(z) \right](t) \qquad\qquad (t > 0) \,, \tag{86}$$

which can be inverted into:

$$U_\mu^{(c)}(z) = \frac{1}{\Gamma(c)} \mathcal{L}_t\left[ t^{c-1} \mathcal{I}_\mu^{(c)}(t) \right](z) \qquad\qquad (\mathfrak{Re}\, z > a_{\max}) \,, \tag{87}$$

and then extended analytically to all $z \in \mathbb{C} \setminus (-\infty, a_{\max})$ .

We emphasize that we have assumed the measure $\mu$ to be compactly supported so the complex integral contour in the inverse Laplace transform of (86) can always be deformed to have the branch cut on the left side of the integral contour and hence (86) is well defined. If we consider a measure $\mu$ with unbounded support, the inverse Laplace transform is not necessarily well-defined and equation (86) only makes sense as an equality between formal series. In some cases we can use a trick similar

---

[6]One may notice that crossing the branch cut at a point $x_0 < a_{\min}$ introduces a phase $e^{2i\pi c}$, so that when $c$ is an integer one can extend analytically the function to $\mathbb{C} \setminus I$.

to a Wick rotation, namely multiply the argument $z$ by a constant using scaling properties implied by (80) and (82):

$$\mathcal{I}_\mu^{(c)}(t) \to \mathcal{I}_\mu^{(c)}(Kt) \Rightarrow U_\mu^{(c)}(z) \to K^{-c} U_\mu^{(c)}\left(\frac{z}{K}\right). \tag{88}$$

If by such a scaling the formal power series now converge, the rescaled functions are then equal on their domain of convergence. In particular, if we look at a measure whose support is of the type $(a, \infty)$, then taking $K = -1$ amounts to look at the measure $\mu(-.)$ whose support is $(-\infty, -a)$ which makes the inverse Laplace transform converges. This is reminiscent of the fact that for measures on $\mathbb{R}_+$, the Laplace transform is more appropriate analytically than the moment generating function.

Following the derivation of Section 2.2.2 together with (83), we have again:

$$U_\mu^{(c)}(z) = \mathcal{I}_\mu^{(c)}(-\mathrm{D})\, z^{-c}. \tag{89}$$

To establish the high temperature counterpart of (35), one can first fix $\beta_{(N)} = \frac{2c}{N}$ and a corresponding sequence of finite measure $\mu_{\boldsymbol{a}_{(N)}}$ such that $\mu_{\boldsymbol{a}_{(N)}} \to \mu$ and $a_{\min(N)}$ and $a_{\max(N)}$ converge towards the edge of the support $I$, and then simply take the limit $N \to \infty$ in (35) accordingly, so that we have:

$$\mathcal{I}_\mu^{(c)}(t) = \mathbb{E}_{X \sim \mathcal{M}_{c,\mu}}\left[e^{tX}\right], \tag{90}$$

where the measure $\mathcal{M}_{c,\mu}$ is known as the *Markov-Krein Transform* (MKT) of $\mu$. The MKT is discussed in great details in [43], where the link with RMT is made. Although not explicit and studied in the regime ($\beta > 0$) instead of the high temperature regime, the link with the HCIZ integral can be directly derived from results of [44].

## 4.2 Generalized Stieltjes transform and fractional calculus

To have a better understanding of the properties of the MKT we first need to introduce the *generalized Stieltjes transform* which is the purpose of this section. For a compactly supported measure $\nu$ with support $J$ with left and right extremities $b_{\min}$ and $b_{\max}$, and $s > 0$, the *generalized Stieltjes transform of order $s$* is defined for all $z \in \mathbb{C} \setminus (-\infty, b_{\max})$ [7] by:

$$\mathcal{G}_\nu^{(s)}(z) := \int_J dx\, \frac{\nu(x)}{(z-x)^s}. \tag{91}$$

For $s = 1$, we drop the superscript and write simply $\mathcal{G}_\nu(.)$ as one recovers the usual definition of the Stieltjes transform. Taking the Taylor expansion of the power function, one arrives at the following formal expansion for the generalized Stieltjes transform:

$$\mathcal{G}_\nu^{(s)}(z) = \frac{1}{\Gamma(s)} \sum_{k=0}^{\infty} \frac{\Gamma(s+k)}{k!} m_k(\nu) z^{-k-s}, \tag{92}$$

where $m_k(\nu)$ is the $k^{th}$ moment of the measure $\nu$, with the usual convention $m_0(\nu) = 1$. It worth noting that using:

$$\frac{1}{(z-x)^s} = \frac{1}{\Gamma(s)} \int_0^\infty dt\, t^{s-1} e^{-t(z-x)}, \tag{93}$$

---

[7]For $s$ integer, one can extend the function to $\mathbb{C} \setminus J$

we can rewrite:

$$\mathcal{G}_\nu^{(s)}(z) := \frac{1}{\Gamma(s)} \mathcal{L}_t \left[ t^{s-1} \mathbb{E}_{X \sim \nu} \left[ e^{tX} \right] \right] (z) . \tag{94}$$

For a measure defined on $\mathbb{R}_+$ and $s = 1$, we recover the fact that up to a sign, the Stieltjes transform is an iterated Laplace transform:

$$-\mathcal{G}_\nu(-z) = \mathcal{L}_t \left[ \mathcal{L}_x \left[ \nu(x) \right] (t) \right] (z) . \tag{95}$$

To connect the generalized and standard Stieltjes transforms, observed that for $s > 1$, one has:

$$x^{-s} = \frac{1}{\Gamma(s)} \mathrm{D}^{s-1} x^{-1} , \tag{96}$$

where for $0 < s < 1$, $\mathrm{D}^{s-1} = \mathrm{D}^{-(1-s)}$ is the fractional anti-derivative[8] of order $\alpha = 1 - s$, defined by:

$$\mathrm{D}^{-\alpha} f(x) := \frac{1}{\Gamma(\alpha)} \int_x^\infty dy \, (y - x)^{\alpha-1} f(y) , \tag{97}$$

and for $s > 1$, it is the fractional derivative of order $\alpha = s - 1$:

$$\mathrm{D}^\alpha f(x) := \mathrm{D}^{\alpha - \lfloor \alpha \rfloor} \mathrm{D}^{\lfloor \alpha \rfloor} f(x) := -\frac{1}{\Gamma(\lfloor \alpha \rfloor + 1 - \alpha)} \int_x^\infty dy \, (y - x)^{-\alpha} \frac{d^{\lfloor \alpha \rfloor + 1}}{dy^{\lfloor \alpha \rfloor + 1}} f(y) , \tag{98}$$

for $\alpha \in \mathbb{N}_+$, the fractional derivative is the usual derivative (by analytical continuation in $\alpha$) and we have in the general case the identity:

$$\mathrm{D}^\alpha \mathrm{D}^{-\alpha} = \mathrm{D}^0 = \mathrm{Id} , \tag{99}$$

Then we have:

$$\mathcal{G}_\nu^{(s)}(z) = \frac{1}{\Gamma(s)} \mathrm{D}^{s-1} \mathcal{G}_\nu(z) . \tag{100}$$

It will useful later on to develop an inverse formula similar to the famous Plemelj inversion formula in the $s = 1$ case:

$$\nu(x) = \frac{1}{\pi} \mathfrak{Im} \lim_{\eta \searrow 0} \mathcal{G}_\nu \left( x - i\eta \right) . \tag{101}$$

If we denote by:

$$g^{(s)}(x) := \frac{1}{\pi} \mathfrak{Im} \lim_{\eta \searrow 0} \mathcal{G}_\nu^{(s)} \left( x - i\eta \right) , \tag{102}$$

then taking the corresponding limit in (100) together with (101) and using the identity (99) yields:

$$\nu(x) = \Gamma(s) \, \mathrm{D}^{1-s} \, g^{(s)}(x) , \tag{103}$$

which gives explicitly:

---

[8]Note that we are interested in functions that are regular at infinity but not necessarily near zero, hence we integrate to infinity and not from zero asit is more customary.

- For $0 < s < 1$:

$$\nu(x) := -\int_x^\infty dy\,(y-x)^{s-1}\frac{d}{dy}g^{(s)}(y)\,, \tag{104}$$

- for $s > 1$:

$$\nu(x) = (s-1)\int_x^\infty dy\,(y-x)^{s-2}g^{(s)}(y)\,. \tag{105}$$

## 4.3 Properties of the Markov Krein transform

Taking (87) together with (90) and (94) we have that the MKT is linked to the original measure by:

$$\mathcal{G}^{(c)}_{\mathcal{M}_{c,\mu}}(z) = U^{(c)}_\mu(z)\,. \tag{106}$$

This relation and its application to different fields is explained in Kerov [43]. More explicitly this writes:

$$\int_J dx\,\frac{\mathcal{M}_{c,\mu}(x)}{(z-x)^c} = \exp\left\{-c\int_I dx\,\log(z-x)\mu(x)\right\}\,, \tag{107}$$

which can also be seen as a non linear differential equation:

$$\frac{d}{dz}\mathcal{G}^{(c)}_{\mathcal{M}_{c,\mu}}(z) + c\,\mathcal{G}^{(c)}_{\mathcal{M}_{c,\mu}}(z)\,\mathcal{G}_\mu(z) = 0\,. \tag{108}$$

Since the functions on the LHS and RHS of (107) are equal and analytical on the complex plane except for the real line going from $-\infty$ to the right extremity of the support of their respective distribution and since for both function crossing the branch cut on the left of the support simply introduces a phase $e^{2\mathrm{i}\pi c}$, we have necessarily equality between the support of the two distributions.

Next using the formal series expansions (83) and (92) together with the definition of the normalized Jack polynomials in the high temperature regime (80), we can express the moments of the MKT $m_k\left(\mathcal{M}_{c,\mu}\right) = \int dx\,x^k \mathcal{M}_{c,\mu}(x)$ in terms of the moments $m_k(\mu) = \int dx\,x^k\mu(x)$ of the original measure:

$$m_k\left(\mathcal{M}_{c,\mu}\right) = \frac{\Gamma(c)\,k!}{\Gamma(c+k)}\sum_{1j_1+\cdots+kj_k=k}c^{j_1+\cdots+j_k}\prod_{i=1}^k\frac{m_i(\mu)^{j_i}}{i^{j_i}j_i!}\,. \tag{109}$$

For completeness we give the inverse mapping, together with (84) and (92) at $s=1$, we have:

$$m_k(\mu) = \frac{k}{c}\sum_{1j_1+\cdots+kj_k=k}(-1)^{\sum_i j_i-1}\left(\sum_i j_i - 1\right)!\prod_i\left(\frac{\Gamma(c+i)}{\Gamma(c)i!}\right)^{j_i}\frac{m_i\left(\mathcal{M}_{c,\mu}\right)^{j_i}}{j_i!}\,. \tag{110}$$

In particular, we have that the means $m_1$ of the two distributions are equal and the variances are linked by:

$$m_2\left(\mathcal{M}_{c,\mu}\right) - m_1\left(\mathcal{M}_{c,\mu}\right)^2 = \frac{1}{c+1}\left(m_2(\mu) - m_1(\mu)^2\right)\,. \tag{111}$$

As in the discrete case, the high temperature MKT has a smaller variance than the original distribution. It is still non zero in this formally $N \to \infty$ regime. In the limit $c \to \infty$ we recover the zero variance found for fixed $\beta$ and infinite $N$. From (107), one can show that a shift and a scaling applied to a density introduces the same shift and scaling to its MKT. Similarly, for an original distribution $\mu$ symmetric, up to a shift we can fix the axis of symmetry to be the $x = 0$ axis without loss generality, then we have that the RHS of (107) is invariant under the symmetry $z \to -z$ and therefore so does the LHS, which implies that the MKT is also symmetric along the same axis. It turns out that if we assume furthermore the distribution $\mu$ to be *unimodal* (in addition to being symmetric), than its MKT is also unimodal with the same vertex [45] but unlike the previous properties, the converse is not true.

Using the Taylor expansion in $c$ of the exponential function in (107), we have:

$$1 - c \int_{\mathbb{R}} dx \log(z - x) \mathcal{M}_{c,\mu}(x) + O\left(c^2\right) = 1 - c \int_{\mathbb{R}} dx \log(z - x) \mu(x) + O\left(c^2\right) , \tag{112}$$

from which we derive the limit:

$$\lim_{c \to 0} \mathcal{M}_{c,\mu} \to \mu . \tag{113}$$

Similarly taking the limit $c \to \infty$ in (111) we immediately find the other extreme case:

$$\lim_{c \to \infty} \mathcal{M}_{c,\mu} \to \delta(x - m_1) . \tag{114}$$

We now aim at finding an explicit expression for the distribution of the MKT. Taking the imaginary part of the RHS of (107) in the limit $\eta \searrow 0$ together with $z = x - i\eta$ , $x \in I$ and the behavior of the logarithm near the real axis, one can derive the following limit [44]:

$$g^{(c)}(x) = \frac{1}{\pi} e^{-c \int dy \, \log|x-y| \mu_A(y)} \sin\left(\pi c \mu_A[x, \infty]\right) \qquad (x \in I) . \tag{115}$$

From which we get the density of the MKT with the proper inversion formula (104) for $c < 1$ and (105) for $c > 1$, while for $c = 1$, we have directly $\mathcal{M}_{1,\mu}(.) = g^{(1)}(.)$. In particular we have that the MKT density is absolutely continuous with respect to the Lebesgue measure.

Next we give few examples of the MKT of distribution that have already appeared before in the literature and that will be useful later on.

### 4.4 Known Markov Krein transforms

**MKT of the Bernoulli distribution:** Let us denote by

$$\mu_{B(p)}(x) := (1 - p)\delta(x - 0) + p\delta(x - 1) , \tag{116}$$

the *Bernoulli distribution* with probability of success $p$, then one can show [46] [45] that its MKT follows the law of a *beta distribution* $\beta(cp, c(1-p))$ so that we have:

$$\mathcal{M}_{c,\mu_{B(p)}}(x) = \frac{\Gamma(c)}{\Gamma(cp)\,\Gamma(c(1-p))} x^{cp-1} (1 - x)^{c(1-p)-1} \, \mathbb{I}_{[0,1]} , \tag{117}$$

where $\mathbb{I}$ is the indicator function. It is worth mentioning that the result can be derived by first looking at the finite setting case and then take the high temperature limit. The vector $\boldsymbol{a} = (0, \ldots, 0, 1, \ldots, 1)$ of size $N$ with $pN$ non-zero values equal to 1 has a spectral distribution given by (116). By the symmetry (12), we can re-scale $\beta$ and $N$ by $pN$ accordingly so that the computation of the corresponding HCIZ integral boils down to the computation of a rank one normalized Jack polynomial which is given by [32]:

$$g_k^{\left(\frac{2}{\beta}\right)}(1, 0, \ldots, 0) = \frac{\prod_{i=0}^{k-1}\left(\frac{\tilde{\beta}}{2} + i\right)}{k!} , \tag{118}$$

Taking the high temperature regime in (9), we get that the $c$-HCIZ is given by:

$$\mathcal{I}^{(c)}_{\mu_{B(p)}}(t) = {}_1F_1(cp, c, t) , \tag{119}$$

which is the moment generating function of (117).

**MKT of the arcsine distribution:**   Another known example in closed form (see for example [45] and reference therein) is given when the original distribution is the *arcsine distribution*:

$$\mu_{As}(x) := \frac{1}{\pi\sqrt{x(1-x)}}\mathbb{I}_{[0,1]} , \tag{120}$$

then one may show that its MKT follows the law of a beta distribution $\beta(c + \frac{1}{2}, c + \frac{1}{2})$:

$$\mathcal{M}_{c,\mu_{As}}(x) = \frac{\Gamma(2c+1)}{\Gamma\left(c + \frac{1}{2}\right)^2}\left(x(1-x)\right)^{c-\frac{1}{2}}\mathbb{I}_{[0,1]} , \tag{121}$$

this can be checked by computing the LHS and RHS of (107) with the corresponding measures.

**MKT of the uniform distribution:**   If we now take the original distribution to be the uniform distribution on $[0, 1]$:

$$\mu_U := \mathbb{I}_{[0,1]} , \tag{122}$$

then by (115) we have:

$$g^{(c)}(x) = \frac{e^c}{\pi}(1-x)^{-c(1-x)}x^{-cx}\sin\left(\pi c(1-x)\right)\mathbb{I}_{[0,1]} , \tag{123}$$

which gives in particular the density for $c = 1$ of the corresponding MKT transform. For $c < 1$ and $c > 1$, one needs to use the formula (104) and (105) but no analytical expression is known.

**MKT of the Cauchy distribution:**   For every $c > 0$, the Markov-Krein transform of a *Cauchy distribution* with parameters $x_0$ and $b$:

$$\mu_{C_{x,b}}(x) := \frac{b}{\pi\left(b^2 + (x - x_0)^2\right)} , \tag{124}$$

is again a Cauchy distribution with the same parameters (which can be seen by computing LHS and RHS of (107), see for example [44] [47]).

## 4.5 Inverting the Markov-Krein transform

For a given measure $\mu$ and a positive real $c$, we have seen that there is always a unique well-defined probability measure which is its MKT. It is natural to ask the reverse question: for a probability measure $\nu$ and a positive real $c$, can we find and express a measure $\mu$ such that $\nu$ is the MKT of $\mu$? The measure $\mu$ will therefore be the *inverse Markov-Krein Transform* (IMKT) of $\nu$ and denoted by $\mathcal{M}_{c,\nu}^{-1}$. Kerov proved that the IMKT of a probability measure always exists and is unique but not necessarily positive and characterized more generally the image of the set of probability measure by the IMKT and we refer to [43] for more details. We aim now at expressing the measure of the IMKT given the density $\nu(.)$.

From (85) and (107), one has:

$$\mathcal{M}_{c,\nu}^{-1}(x) = -\frac{1}{c\pi}\frac{d}{dx}\lim_{\eta\searrow 0}\mathfrak{Im}\log\left\{\mathcal{G}_\nu^{(c)}(x-i\eta)\right\}, \tag{125}$$

$$\mathcal{M}_{c,\nu}^{-1}(x) = -\frac{1}{c\pi}\frac{d}{dx}\arctan\left(\frac{\lim_{\eta\searrow 0}\mathfrak{Im}\mathcal{G}_\nu^{(c)}(x-i\eta)}{\lim_{\eta\searrow 0}\mathfrak{Re}\mathcal{G}_\nu^{(c)}(x-i\eta)}\right), \tag{126}$$

where we have used that imaginary part of the logarithm is (up to an irrelevant constant) the *arctan* function of the ratio of the imaginary and real part of its argument. If we know the generalized Stieltjes of $\nu$, (126) can be used directly. We can also use the link between the generalized and the standard Stieltjes transform via the fractional derivative to express the IMKT measure more directly as a function of $\nu$. From Section 4.2, we already know that:

$$\lim_{\eta\searrow 0}\mathfrak{Im}\,\mathcal{G}_\nu^{(c)}(x-i\eta) = \frac{\pi}{\Gamma(c)}\mathrm{D}^{c-1}\,\nu(x). \tag{127}$$

similarly for the real part, since for $c = 1$ we have:

$$\frac{1}{\pi}\lim_{\eta\searrow 0}\mathfrak{Re}\,\mathcal{G}_\nu(x-i\eta) = \mathcal{H}_\nu(x), \tag{128}$$

where

$$\mathcal{H}_\nu(x) := \frac{1}{\pi}\mathrm{P.V}\int_I dy\,\frac{\nu(y)}{x-y}, \tag{129}$$

is the *Hilbert transform* of the measure $\nu$ and P.V indicates that the integral has to be understood as a Cauchy principal value integral, we find:

$$\lim_{\eta\searrow 0}\mathfrak{Re}\,\mathcal{G}_\nu^{(c)}(x-i\eta) = \frac{\pi}{\Gamma(c)}\mathrm{D}^{c-1}\,\mathcal{H}_\nu(x). \tag{130}$$

Equation (126) can therefore be written as:

$$\mathcal{M}_{c,\nu}^{-1}(x) = -\frac{1}{c\pi}\frac{d}{dx}\arctan\left(\frac{\mathrm{D}^{c-1}\,\nu(x)}{\mathcal{H}_{\mathrm{D}^{c-1}\,\nu}(x)}\right), \tag{131}$$

where we have used the fact the the fractional derivative and the Hilbert transform are both linear kernel operators and therefore commute. If the density $\mathcal{M}_{c,\nu}^{-1}$ is continuous, this reads:

$$\mathcal{M}_{c,\nu}^{-1}(x) = \frac{1}{c\pi}\frac{\mathcal{H}_{\mathrm{D}^c\,\nu}(x)\,\mathrm{D}^{c-1}\,\nu(x) - \mathcal{H}_{\mathrm{D}^{c-1}\,\nu}(x)\,\mathrm{D}^c\,\nu(x)}{(\mathcal{H}_{\mathrm{D}^{c-1}\,\nu}(x))^2 + (\mathrm{D}^{c-1}\,\nu(x))^2}. \tag{132}$$

One has to be careful when applying (131) or (132), while the density of the IMKT is defined on the same support as that of the measure $\nu$, the fractional derivative is a non local operator and should be compute for all $x < a_{\max}$ before computing its Hilbert transform. Note as well that our definition of fractional derivative uses a boundary condition at infinity rather than the more usual boundary at zero. For these reasons these formulas are difficult to use in practice; except for integer $c$ where the fractional derivative reduces to the usual derivative.

We finish this section with several examples of IMKT:

**IMKT of the standard Gaussian distribution:**   For

$$\nu_G(x) := \frac{e^{-\frac{x^2}{2}}}{\sqrt{2\pi}} \,, \tag{133}$$

a *standard Gaussian distribution*, then one has that the IMKT is given by the so-called *Askey-Wimp-Kerov distribution*:

$$\mathcal{M}_{c,\nu_G}^{-1}(x) = \frac{1}{\sqrt{2\pi}\Gamma(c+1)} \frac{1}{|D_{-c}(\mathrm{i}x)|^2} \,, \tag{134}$$

where $D_{-c}(.)$ is a *parabolic cylinder function* defined by:

$$D_{-c}(z) := \frac{e^{-\frac{z^2}{4}}}{\Gamma(c)} \int_0^\infty dx \, e^{-zx-\frac{x^2}{2}} x^{c-1} \,, \tag{135}$$

This fact was first obtained by Kerov [43], while the distribution had first appeared in [48] as the distribution whose orthogonal polynomials are the *associated Hermite polynomials*, since then it has appeared also in RMT ensemble at high temperature [12] [14]. This distribution is a continuous interpolation between the Gaussian distribution at $c = 0$ and the unnormalized (with infinite variance) semi-circle distribution at $c \to \infty$.

**IMKT of the gamma distribution:**   The *gamma distribution* with parameter $(k, \theta)$ is defined by the probability density:

$$\nu_{\gamma(k,\theta)}(x) := \frac{e^{-\frac{x}{\theta}} x^{k-1}}{\Gamma(k)\,\theta^k} \,, \tag{136}$$

Since the parameter $\theta$ is a scale parameter, we can fix it to $\theta = 1$ without loss of generality thanks to the scaling property of Section 4.3 and we simply denote by $\nu_{\gamma(k)}$ the gamma distribution in this case. Since the support of the measure is $(0, \infty)$, it will be more convenient to characterize it by its Laplace transform:

$$\mathbb{E}_{X \sim \gamma(k)}\left[e^{-tX}\right] = (1+t)^{-k} \,, \tag{137}$$

then using the following identity for the *Tricomi function* $\Psi(.)$:

$$\Psi(c, c+1-k; z) := \frac{1}{\Gamma(c)} \int_0^\infty dt \, e^{-zt} t^{c-1}(1+t)^{-k} \qquad (\mathfrak{Re}\, z > 0) \tag{138}$$

Taking care of the branch cut on the negative real axis of the Tricomi function and using property (88), we have for $z \in \mathbb{C} \setminus \mathbb{R}_+$, that up to a multiplicative constant that is irrelevant:

$$U_{\mathcal{M}_{c,\nu}^{-1}}^{(c)}(z) \propto \Psi(c, c+1-k; -z) \,. \tag{139}$$

Since we have:

$$\frac{d}{dz}\Psi\left(c, c+1-k; -z\right) = c\,\Psi\left(c+1, c+2-k; -z\right) . \tag{140}$$

Using (84) we find that the corresponding Stieltjes transform is given by:

$$\mathcal{G}_{\mathcal{M}_{c,\nu}^{-1}}(z) = -\frac{\Psi\left(c+1, c+2-k; -z\right)}{\Psi\left(c, c+1-k; -z\right)} . \tag{141}$$

It turns out that a similar Stieltjes transform had already appear in the RMT literature in a different context [13] [17] from which we can immediately get the limiting density:

$$\mathcal{M}_{c,\nu}^{-1}(x) = \frac{1}{\Gamma(c+1)\Gamma(k)} \frac{x^{k-c-1}e^{-x}}{\left|\Psi\left(c, c+1-k; e^{i\pi^-} x\right)\right|^2} \mathbb{I}_{x>0} + \Theta(c-k)\frac{c-k}{c}\delta(x-0) . \tag{142}$$

where $\Theta(.)$ is the Heaviside step function which is equal to 0 for $x > 0$ and 1 otherwise. Crossing the branch cut of the Tricomi function introduces a change in the sign of the imaginary part of the function so that the function $|\Psi\left(\alpha_1, \alpha_2; .\right)|$ can be continued analytically to all $\mathbb{C}$. The density in (142) does not depend on the choice of the branch cut, in particular one could have taken instead $e^{i\pi^+}$ in the argument of the Tricomi function. The case $\theta \neq 1$ is then obtained by dilatation and we have:

$$\mathcal{M}_{c,\nu}^{-1}(x) = \frac{\theta^{c-k}}{\Gamma(c+1)\Gamma(k)} \frac{x^{k-c-1}e^{-\frac{x}{\theta}}}{\left|\Psi\left(c, c+1-k; -\frac{x}{\theta}\right)\right|^2} \mathbb{I}_{x>0} + \Theta(c-k)\frac{c-k}{c}\delta(x-0) . \tag{143}$$

This distribution has mean $k\theta$ and variance $k\theta^2(c+1)$, it interpolates between the gamma distribution (at $c = 0$) and the (rescaled) Marčenko-Pastur distribution. To recover the standard Marčenko-Pastur with aspect ratio $q$, one has to take the limit $c \to \infty$ with $k \to qc$ and $\theta \to (qc)^{-1}$.

**IMKT of the beta distribution:** It is natural to ask if one can find a positive measure for the IMKT of the beta distribution since it is the third classical ensemble after the Gaussian and the gamma distribution of the two previous examples. We will actually show the opposite by finding a triplet $(c, a, b)$ where $(a, b)$ are the parameters of the beta distribution, such that the IMKT is not positive. Since we have that the moment generating function of the beta distribution $\beta(a, b)$ is given by:

$$\mathbb{E}_{X \sim \beta(a,b)}\left[e^{tX}\right] = {}_1F_1\left(a, a+b; t\right) . \tag{144}$$

We have from (87) and (90) that the corresponding IMKT satisfies:

$$U_{\mathcal{M}_{c,\nu}^{-1}}^{(c)}(z) = \frac{1}{\Gamma(c)} \int_0^\infty dt\, e^{-zt} t^{c-1} {}_1F_1\left(a, a+b; t\right) , \tag{145}$$

by the classical identity between the hypergeometric functions:

$${}_2F_1\left(a, c, a+b; z\right) = \frac{1}{\Gamma(c)} \int_0^\infty dt\, e^{-t} t^{c-1} {}_1F_1\left(a, a+b; zt\right) , \tag{146}$$

we find:

$$U_{\mathcal{M}_{c,\nu}^{-1}}^{(c)}(z) = z^{-c}\, {}_2F_1\left(a, c, a+b; 1/z\right) . \tag{147}$$

From (84) together with the identity for the derivative of the hypergeometric function ${}_2F_1(.)$:

$$\frac{d}{dx}\, _2F_1(\alpha,\beta,\gamma;x) = \frac{\alpha\beta}{\gamma}\, _2F_1(\alpha+1,\beta+1,\gamma+1;x)\,,\tag{148}$$

we get after simplification

$$\mathcal{G}_{\mathcal{M}_{c,\nu}^{-1}}(z) = \frac{1}{z} + \frac{a\, _2F_1\left(a+1,c+1,a+b+1;1/z\right)}{(a+b)z^2\, _2F_1\left(a,c,a+b;1/z\right)}\,.\tag{149}$$

This expression is the Stieltjes transform of the IMKT of a beta distribution with arbitrary parameters $a$ and $b$, in particular we recover the Stieltjes of the Bernouilli (116) for $a=cp$ and $b=c(1-p)$ and the arcsine law (120) for $a=b=c+1/2$.

As an explicit example of a non positive IMKT we fix $c=2$, $a=b=\frac{1}{2}$, in this case the expression simplifies considerably and we have:

$$\mathcal{G}_{\mathcal{M}_{c,\nu}^{-1}}(z) = \frac{3-8z+8z^2}{4z-12z^2+8z^3}\,,\tag{150}$$

$$\mathcal{G}_{\mathcal{M}_{c,\nu}^{-1}}(z) = \frac{3}{4}\frac{1}{z-1} + \frac{3}{4}\frac{1}{z} - \frac{1}{2z-1}\,.\tag{151}$$

From which we derive the corresponding measure is the discrete measure:

$$\mathcal{M}_{c,\nu}^{-1}(x) = \frac{3}{4}\delta(x-0) - \frac{1}{2}\delta\left(x-\frac{1}{2}\right) + \frac{3}{4}\delta(x-1)\,,\tag{152}$$

and hence it is not positive.

# 5  $c$-convolution

## 5.1  $c$-convolution as convolution of Markov-Krein transforms

Since the HCIZ integral is multiplicative for the free convolution for $\beta>0$, in the limit $N\to\infty$, and multiplicative for the classical convolution at $\beta=0$, it is natural to construct a new convolution, which we call the *c-convolution* and denote it by $\oplus_c$, for which the HCIZ in the high temperature regime $\frac{N\beta}{2}\to c$ of the previous section is multiplicative. Using (90) this is equivalent to say that our *c-convolution* corresponds to a classical convolution in the *Markov-Krein* space. This statement can be summarized by the following scheme:

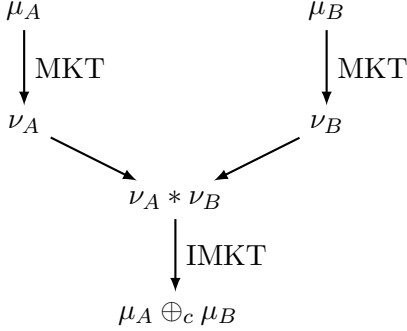

The $c$-convolution

- is commutative: $\mu_A \oplus_c \mu_B = \mu_B \oplus_c \mu_A$ ,

- is associative: $\mu_A \oplus_c (\mu_B \oplus_c \mu_C) = (\mu_A \oplus_c \mu_B) \oplus_c \mu_C$ ,

- is well behaved with respect to shift: $\mu_A \oplus_c \delta_{x_0} = \mu_A(. - x_0)$ ,

- preserves symmetric measures: if two distributions are symmetric with respect to their means than their $c$-convolution is also symmetric wrt its mean,

- is additive for the means and the variances:

$$m_2\left(\mu_A \oplus_c \mu_B\right) - \left(m_1(\mu_A \oplus_c \mu_B)\right)^2 = m_2\left(\mu_A\right) - m_1(\mu_A)^2 + m_2\left(\mu_B\right) - m_1(\mu_B)^2 , \tag{153}$$

- admits the limits $\lim_{c \to 0} \mu_A \oplus_c \mu_B = \mu_A * \mu_B$ and $\lim_{c \to \infty} \mu_A \oplus_c \mu_B = \mu_A \boxplus \mu_B$.

All of this properties are derived immediately from the properties of the Markov-Krein of Section 4.3. In the general setting, the $c$-convolution is defined on the set of the image of the IMKT described by Kerov [43] (see also [44]), it is an open and important question to know whether the $c$-convolution is stable for probability measures.

We emphasis that this convolution is well suited for numerical simulations since the operations to compute the MKT on the one hand, namely (115) together with (104) or (105) and the ones to compute the IMKT with (132) or with (125) and the definition (91) can all be approximate numerically. We have illustrated the results of the $c$-convolution of several well known examples of distribution in the classical and free world in Fig. 1.

## 5.2 $c$-cumulants

The $c$-convolution being defined, the next step is to define the corresponding $c$-cumulants which we denote by $\kappa_k^{(c)}$. Following Lehner [49], the $c$-cumulants must satisfy

- additivity: $\kappa_k^{(c)}\left(\mu_A \oplus_c \mu_B\right) = \kappa_k^{(c)}\left(\mu_A\right) + \kappa_k^{(c)}\left(\mu_B\right)$ ,

- homogeneity: $\kappa_k^{(c)}\left(\frac{1}{\lambda}\mu_A\left(\frac{.}{\lambda}\right)\right) = \lambda^k \kappa_k^{(c)}\left(\mu_A\right)$ ,

- $\kappa_k^{(c)}$ is a polynomial in the first $k$ moments with leading term $m_k$.

By construction of the $c$-convolution we have that the (classical) cumulants of the MKT are additive (and of course homogeneous) for the $c$-convolution, but their leading term is given by the $k^{th}$ moment of the MKT and not the $k^{th}$ moment of the original distribution, so that we need to compute the leading term $C_{k,c}$ in the development:

$$m_k\left(\mathcal{M}_{c,\mu}\right) = C_{k,c}\, m_k\left(\mu\right) + \ldots , \tag{154}$$

which using (109) is given:

$$C_{k,c} = \frac{\Gamma(c+1)(k-1)!}{\Gamma(c+k)} , \tag{155}$$

hence dividing by $C_{k,c}$ the classical cumulant of the MKT with get the $c$-cumulant, from which we derive that they satisfy the following equation:

$$\log\left(1 + \sum_{k=1}^{\infty} \frac{m_k\left(\mathcal{M}_{c,\mu}\right)}{k!} t^k\right) = \sum_{k=1}^{\infty} \frac{\Gamma(c+1)}{\Gamma(c+k)k} \kappa_k^{(c)} t^k . \tag{156}$$

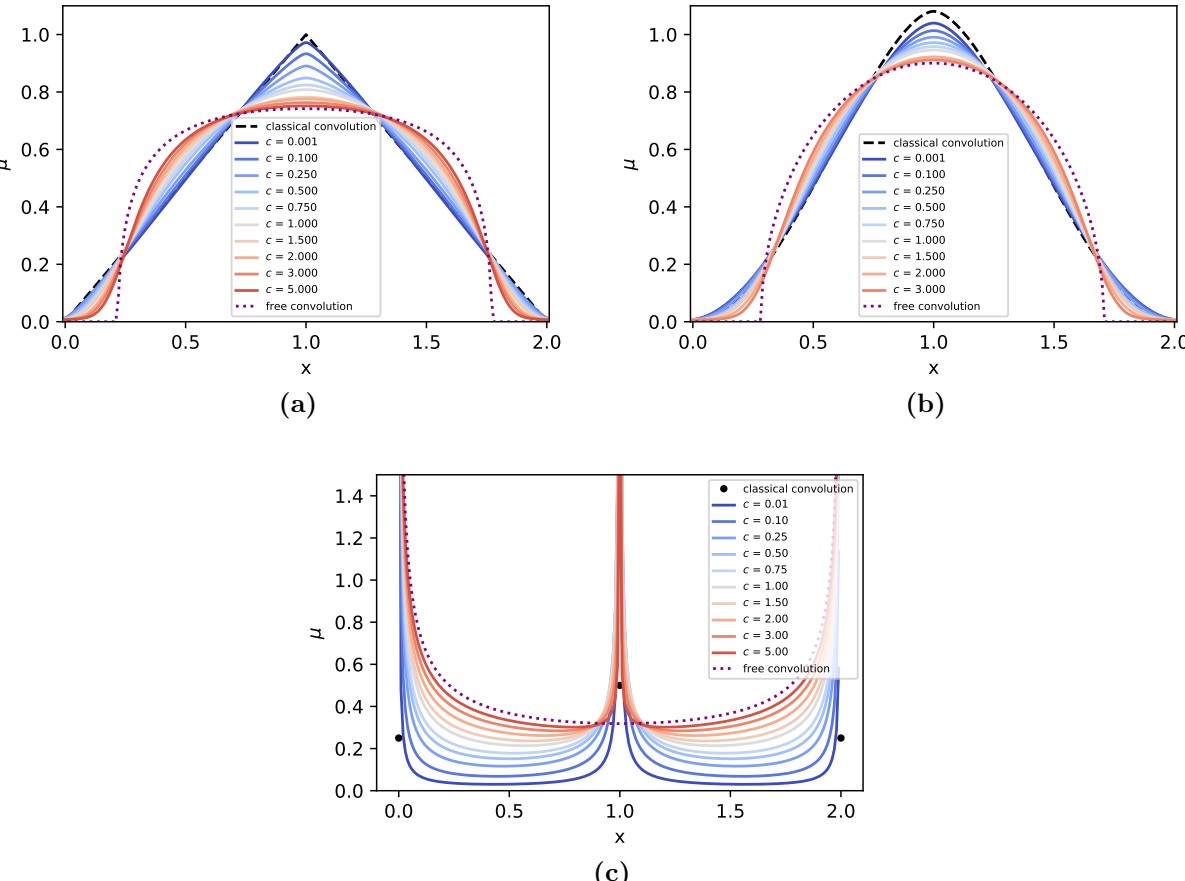

**Figure 1:** Plots of numerical approximations of the c-convolution of the uniform distribution **(a)** the semi-circle **(b)** and the symmetric Bernoulli distribution **(c)** with itself for different values of $c$. The dashed line corresponds to the classical convolution and the dotted line to the free convolution limiting cases.

For completeness we give the cumulant-moment expression:

$$\kappa_k^{(c)} = \frac{\Gamma(c+k)k}{c} \sum_{1j_1+\cdots+kj_k} (-\Gamma(c))^{\sum_i j_i - 1} \frac{(\sum_i j_i - 1)!}{\prod_i j_i!\,\Gamma(c+i)^{j_i}} \prod_i \left( \sum_{1l_1+\cdots+il_i=i} c^{\sum l_n} \prod_n \frac{m_n^{l_n}}{n^{l_n}\,l_n!} \right)^{j_i},$$

(157)

from which we can derive the first cumulants-moment relations:

$$\kappa_1^{(c)} = m_1\,,$$
$$\kappa_2^{(c)} = m_2 - m_1^2\,,$$
$$\kappa_3^{(c)} = m_3 - 3m_2 m_1 + 2m_1^3\,,$$
$$\kappa_4^{(c)} = m_4 - 4m_3 m_1 - \left(2 + \frac{1}{c+1}\right) m_2^2 + \left(10 + \frac{2}{c+1}\right) m_2 m_1^2 - \left(5 + \frac{1}{c+1}\right) m_1^4$$
$$\kappa_5^{(c)} = m_5 - 5m_4 m_1 - 5\left(1 + \frac{1}{c+1}\right) m_3 m_2 + \left(15 + \frac{5}{c+1}\right) m_3 m_1^2 + 15\left(1 + \frac{1}{c+1}\right) m_2^2 m_1$$
$$\qquad - \left(35 + \frac{25}{c+1}\right) m_2 m_1^3 + \left(14 + \frac{10}{c+1}\right) m_1^5\,.$$

In particular when the first moment $m_1 = 0$, we have that the $4^{th}$ cumulant is given by:

$$\kappa_4^{(c)} = m_4 - m_2^2 \left(\frac{2c+3}{c+1}\right)\,,$$

(158)

from which we see that the value $c = 1$ corresponds to the midpoint between the classical and free case.

Similarly we can obtain the moment in terms of terms of the $c$-cumulants, we only give here the first five moment-$c$-cumulant relations:

$$m_1 = \kappa_1^{(c)}\,,$$
$$m_2 = \kappa_2^{(c)} + \left(\kappa_1^{(c)}\right)^2\,,$$
$$m_3 = \kappa_3^{(c)} + 3\kappa_2^{(c)}\kappa_1^{(c)} + \left(\kappa_2^{(c)}\right)^3\,,$$
$$m_4 = \kappa_4^{(c)} + 4\kappa_3^{(c)}\kappa_1^{(c)} + \left(2 + \frac{1}{c+1}\right) \left(\kappa_2^{(c)}\right)^2 + 6\kappa_2^{(c)}\left(\kappa_1^{(c)}\right)^2 + \left(\kappa_1^{(c)}\right)^4\,,$$
$$m_5 = \kappa_5^{(c)} + 5\kappa_4^{(c)}\kappa_1^{(c)} + \left(5 + \frac{5}{c+1}\right) \kappa_3^{(c)}\kappa_2^{(c)} + 10\kappa_3^{(c)}\left(\kappa_1^{(c)}\right)^2 + \left(10 + \frac{5}{c+1}\right) \left(\kappa_2^{(c)}\right)^2 \kappa_1^{(c)}$$
$$\qquad + 10\kappa_2^{(c)}\left(\kappa_1^{(c)}\right)^3 + \left(\kappa_1^{(c)}\right)^5\,.$$

## 5.3 $c$-central limit theorem and related distributions

Let $\mu$ a measure with mean zero and variance one, then we look at the following $c$-Central Limit Theorem ($c$-CLT):

$$\mu_G^{(c)}(.) := \lim_{T \to \infty} \sqrt{T}\mu(\sqrt{T}\,.)^{\oplus_c T}\,,$$

(159)

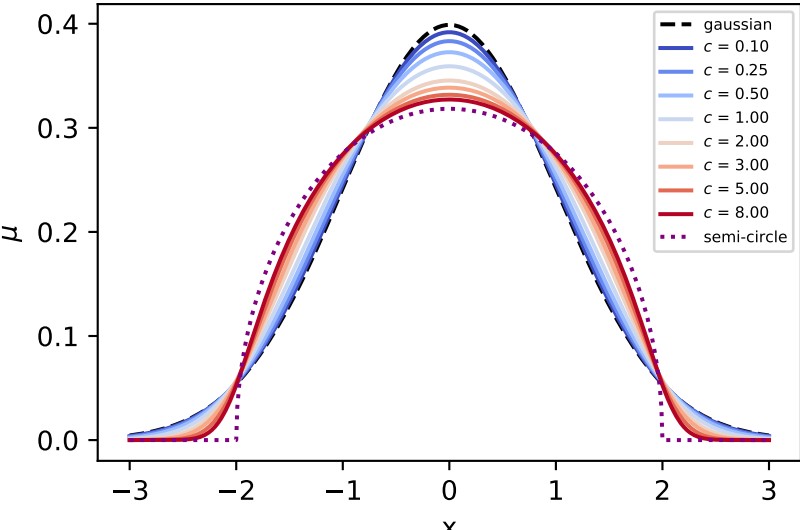

**Figure 2:** Plots of the $c$-Gaussian distribution defined in (160) for different values of $c$, the dashed lined corresponds to the classical limiting case and the doted line the free limiting case.

where $.^{\oplus_c T}$ indicates that we do the $c$-convolution of the measure $\mu$ $T$ times. The $\mu_G^{(c)}(.)$ is the $c$-*Gaussian distribution*, which is the normalized *Askey Wimp Kerov* distribution of equation (134), given by:

$$\mu_G^{(c)}(x) = \frac{\sqrt{c+1}}{\sqrt{2\pi}\Gamma(c+1)} \frac{1}{\left|D_{-c}\left(\mathrm{i}\sqrt{c+1}x\right)\right|^2}. \tag{160}$$

Indeed, the Markov-Krein transform of $\mu_i$ is a distribution with mean zero and variance $\frac{1}{c+1}$, since $c$-convolution corresponds to classical convolution in the Markov-Krein space, we have by the classical central limit theorem that the limiting distribution is the IMKT transform of the Gaussian distribution with variance $\frac{1}{c+1}$. But we known from previous example that the IMKT of the standard Gaussian distribution is given by (134), so by the scaling property derived in Section 4.3 we have the desired result. By construction the orthogonal polynomials of the $c$-Gaussian distribution continuously interpolates between the *Hermite polynomials* of the (classical) Gaussian and the *Chebyshev polynomials* of the second kind of the semi-circle distribution and are known as the (rescaled) *associated Hermite polynomials*, see [48].

As illustrated in Fig 2, this distribution is a continuous interpolation between the standard Gaussian distribution and the semi-circle distribution, in accordance with the properties of the $c$-convolution.

**c-cumulants:** Since the MKT of the $c$-Gaussian is a Gaussian, we find immediately from results of the previous section, that the cumulants of the $c$-Gaussian are defined by:

$$\kappa_k^{(c)} = 1\,\delta_{k,2}, \tag{161}$$

where $\delta_{k,2} = 1$ if $k = 2$ and zero otherwise, which is expected from the limiting distribution of a CLT.

**Infinite divisibility and the gamma Marčenko-Pastur crossover:** In this section, we would like to interpolate between the gamma and Marčenko-Pastur (MP) distribution using their properties under convolution.

We consider the ensemble of gamma distributions (136) parametrized by their mean $k\theta$ and variance $k\theta^2$ and the (scaled) MP distributions of mean $\theta$ and variance $q\theta^2$ defined by:

$$\mu_{MP(q,\theta)}(x) := \left(1 - \frac{1}{q}\right)\delta(x - 0)\Theta(q - 1) + \frac{\sqrt{(x_+ - x)(x - x_-)}}{2\pi q\theta x}, \tag{162}$$

where $\Theta(.)$ is the *Heaviside function* which is equal to 0 for $x < 0$ and 1 otherwise and $x_\pm = \theta(1 \pm \sqrt{q})^2$. The distributions in both ensemble are infinitely divisible (under classical or free convolution respectively) and are closed under scaling and convolution. Multiple families of law satisfy these two conditions, to uniquely determine the gamma and MP distribution we need to specify at least one member of the family: the square-Gaussian (or square semi-circle for MP). Indeed any gamma (MP) can be obtained by scaling, convolution and convolution roots of the square-Gaussian (square-semi-circle), i.e. the random variable $y = x^2$ where $x$ is a unit centered Gaussian (semi-circle), it corresponds to a gamma with $\theta = 2, k = \frac{1}{2}$ (MP with $\theta = 1, q = 1$).

For any $c$, the $c$-gamma distributions given by (143) are infinitely divisible and closed under the $c$-convolution. Indeed, the c-convolution is defined as the convolution of MKTs and the MKT of a c-gamma is a gamma (by construction) themselves infinitely divisible and closed under convolution. For a given mean and variance the $c$-gamma tends to the gamma and Marčenko-Pastur distribution in the limit $c \to 0$ and $c \to \infty$ respectively. Let's see whether the c-gamma family also contains the squared c-Gaussian whose distribution is given by

$$\rho^{(c)}(x) := \frac{1}{\sqrt{x}}\mu_G^{(c)}\left(\sqrt{x}\right) = \frac{\sqrt{c + 1}}{\sqrt{2\pi}\Gamma(c + 1)}\frac{x^{-\frac{1}{2}}}{\left|D_{-c}\left(\mathrm{i}\sqrt{(c + 1)x}\right)\right|^2}, \tag{163}$$

by property of the parabolic cylinder function, we have:

$$D_{-s}(z) = 2^{-s/2}e^{\frac{z^2}{4}}\Psi\left(\frac{s}{2}, \frac{1}{2}; \frac{z^2}{2}\right), \tag{164}$$

Since we are taking the absolute value, we can again extend this formula near the branch cut, from which we have:

$$\rho^{(c)}(x) = \frac{2^c\sqrt{c + 1}}{\sqrt{2\pi}\Gamma(c + 1)}\frac{x^{-\frac{1}{2}}e^{-\frac{c+1}{2}x}}{\left|\Psi\left(\frac{c}{2}, \frac{1}{2}; e^{\mathrm{i}\pi-}\frac{c+1}{2}x\right)\right|^2}\mathbb{I}_{x>0}, \tag{165}$$

where again we could have taken $e^{\mathrm{i}\pi^+}$ in the argument of the Tricomi function without changing the result. We recognize a $\tilde{c}$-gamma distribution (143) with parameters $\tilde{c} = c/2$, $\theta = 2/(c + 1)$ and $k = (c + 1)/2$. The normalizing constants look superficially different but they are indeed equal as it should. Note that $k > \tilde{c}$ so this law doesn't have a mass at zero. The first two moments of both law obviously match and are given by $\mu = 1$ and $\sigma^2 = (\tilde{c} + 1)/(\tilde{c} + 1/2) = (c + 2)/(c + 1)$.

So the $c$-gamma family contains a squared $c$-Gaussian but for a $c$ twice as big. This is still consistent with the $c$-gamma interpolating between the standard gamma and MP distribution as when $c$ goes to either zero or infinity the $2c$-Gaussian and the $c$-Gaussian become identical.

We have plotted in Fig 3, the distribution $\mu_\gamma^{(c)}(.)$ for different values of $c$.

It would be interesting to know if one can construct explicitly a positive measure by replacing the $2c$-Gaussian by the $c$-Gaussian. If this construction exists it would yield a different interpolation between the gamma and the MP than the $c$-gamma considered here.

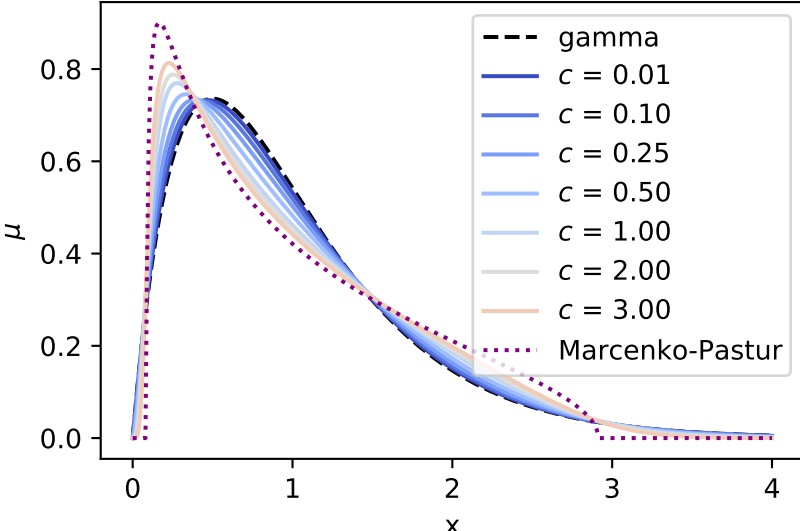

**Figure 3:** Plots of the $c$-gamma with mean 1 and variance $\frac{1}{2}$ for different values of $c$, the dashed lined corresponds to the classical limiting case and the doted line the free limiting case

**$c$-stability of the Cauchy distribution:** By the $c$-CLT, we have that the $c$-Gaussian is $c$-stable, another example of a $c$-stable distribution is given by the Cauchy distribution, since we know that it is a fixed point for both the MKT and the classical convolution, this writes simply:

$$\mu_{C_{x_1,b_1}} \oplus_c \mu_{C_{x_2,b_2}} = \mu_{C_{x_3,b_3}} \,, \tag{166}$$

where the Cauchy distribution is defined in (124) and $x_3 = x_1 + x_2$ and $b_3 = b_1 + b_2$.

### 5.4 $c$-Poisson limit theorem

Another classical limit theorem is the Poisson central limit theorem which concerns limit of sum of independent Bernoulli random variables with a probability of success that goes to zero at a speed $\frac{1}{N}$:

$$\lim_{T\to\infty} \left( \left(1 - \frac{\lambda}{T}\right) \delta(x - 0) + \frac{\lambda}{T}\delta(x - a) \right)^{*T} = \frac{1}{a}\mu_{Poi(\lambda)}\left(\frac{x}{a}\right), \qquad \text{(for } a > 0) \tag{167}$$

where $\mu_{Poi(\lambda)}(x) = \sum_{k=0}^{\infty} e^{-\lambda}\frac{\lambda^k}{k!} \, \delta(x - k)$ is the *Poisson distribution*.

This limit theorem admits a free counterpart:

$$\lim_{T\to\infty} \left( \left(1 - \frac{\lambda}{T}\right) \delta(x - 0) + \frac{\lambda}{T}\delta(x - a) \right)^{\boxplus T} = \frac{1}{a\lambda}\mu_{MP(\frac{1}{\lambda})}\left(\frac{x}{a\lambda}\right) \qquad \text{(for } a > 0)\,, \tag{168}$$

where $\mu_{MP(\lambda)}$ is the *Marčenko-Pastur distribution*, defined in (162).

In this subsection we aim at developing the $c$-counterpart of these theorems, whose limiting objects will interpolate between the Poisson and the re-scaled Marčenko-Pastur distribution. We know from (117), that the Markov-Krein transform of the Bernoulli distribution of probability of

success $p$ is the beta distribution $\beta(cp, c(1-p))$. Since again $c$-convolution corresponds to classical convolution in the MK space, we first need to determine the limiting distribution of:

$$\nu(.) := \lim_{T \to \infty} \left( \frac{1}{a} \beta_{\left( \frac{c\lambda}{T}, \frac{c(T-\lambda)}{T} \right)} \left( \frac{.}{a} \right) \right)^{*T}, \tag{169}$$

and then take the IMKT. This kind of distribution does not seemed to have appeared before in the literature and we will characterize it with its moment generating function (as no analytical solution is known). The moment generating function of the beta distribution is given by (144), so that we have:

$$\mathbb{E}_{X \sim \nu} \left[ e^{tX} \right] = \lim_{T \to \infty} {}_1F_1 \left( \frac{c\lambda}{T}, c; at \right)^T, \tag{170}$$

$$\mathbb{E}_{X \sim \nu} \left[ e^{tX} \right] = \lim_{T \to \infty} \left( 1 + \frac{\sum_{k=1}^{\infty} \frac{\lambda \Gamma(c+1)}{\Gamma(c+k) k} (at)^k}{T} + O\left( \frac{1}{T^2} \right) \right)^T, \tag{171}$$

Next using:

$$_2F_2 \left( \{1, 1\}, \{2, c+1\}; t \right) = \sum_{k=0}^{\infty} \frac{\Gamma(c+1)}{\Gamma(c+k+1)(k+1)} t^k = \frac{1}{t} \sum_{k=1}^{\infty} \frac{\Gamma(c+1)}{\Gamma(c+k) k} t^k, \tag{172}$$

where $_2F_2$ is the *hypergeometric function*. Together with the classical limit identity for the exponential:

$$e^x = \lim_{T \to \infty} \left( 1 + \frac{x}{T} \right)^T, \tag{173}$$

we get:

$$\mathbb{E}_{X \sim \nu} \left[ e^{tX} \right] = \exp \left\{ a\lambda t \, _2F_2 \left( \{1, 1\}, \{2, c+1\}; at \right) \right\}. \tag{174}$$

Since the distribution $\nu$ has support $\mathbb{R}_+$, we can take the inverse Laplace transform of the moment generating function evaluated at $-t$:

$$\nu(x) = \mathcal{L}_t^{-1} \left[ \exp \left\{ -a\lambda t \, _2F_2 \left( \{1, 1\}, \{2, c+1\}; -t \right) \right\} \right] (x). \tag{175}$$

We can therefore compute numerically $\nu$ thanks to (175). We have plotted the distribution for different values in Fig 4.

The $c$-Poisson is then approximate numerically and we have plotted the different result in Fig 5.

**$c$-cumulants:** Using (174) and (155), we have that the $c$-cumulants of the $c$-Poisson are given by:

$$\kappa_k^{(c)} = a^k \lambda. \tag{176}$$

which is again expected from a limiting distribution of a Poisson limit theorem.

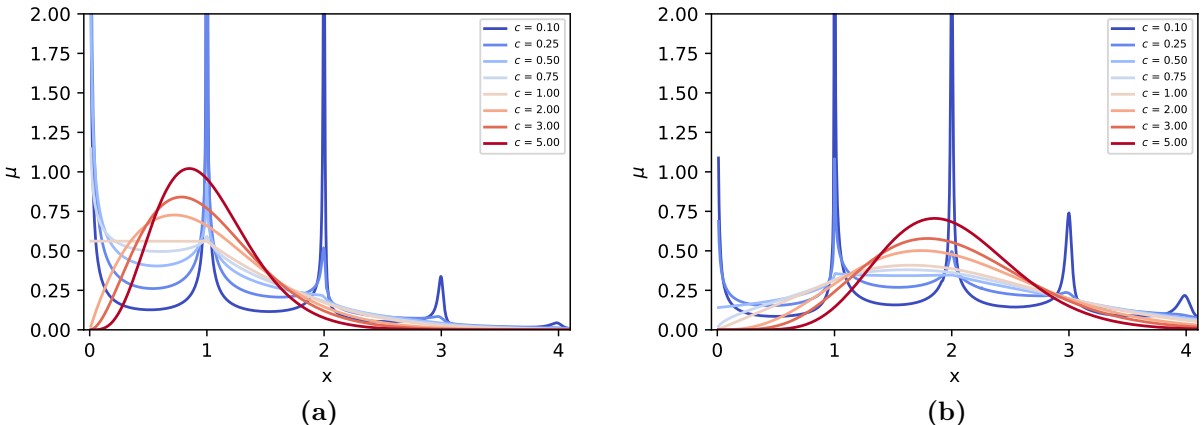

**Figure 4:** Plots of the Markov-Krein transforms of the limiting distributions of the Poisson limit theorem with parameters $a = 1$ , $\lambda = 1$ in (**a**), $\lambda = 2$ in (**b**), for different values of $c$.

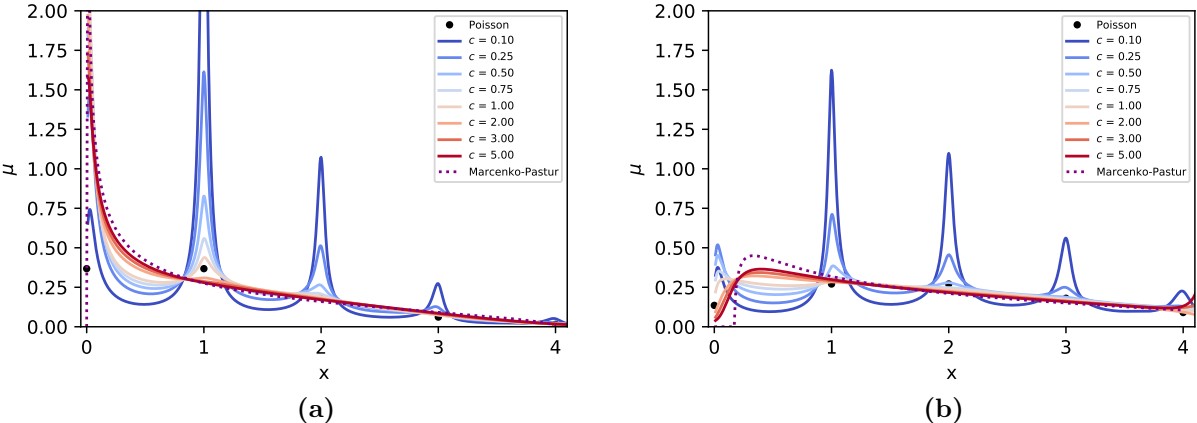

**Figure 5:** Plots of the numerical approximation of the limiting distribution of the Poisson limit theorem with parameters $a = 1$ , $\lambda = 1$ in (**a**), $\lambda = 2$ in (**b**), for different values of $c$. compared to the classical (Poisson) and free (Marčenko-Pastur) limiting distributions.

# 6 Conclusion and open questions

In this note we have construct the $c$-convolution, a one-parameter interpolation between the classical ($c = 0$) and the free ($c \to \infty$) convolution. Our main object of study is the HCIZ integral in the high temperature regime $\frac{N\beta}{2} \to c$, which is multiplicative for this convolution. It turns out that in this regime the HCIZ is the moment generating function of the so-called Markov-Krein transform of the distribution of interest so that the $c$-convolution of two distributions corresponds to a classical convolution of their Markov-Krein transforms. We finish this note with remarks and open questions that we believe are worth mentioning:

- We have not proved that the $c$-convolution preserved positivity and it is therefore possible that the $c$-convolution of two probability distributions is not a probability distribution. More generally, it will be interesting to know if one can (for a given $c$ or better independently of $c$) restrict the set of probability distributions so that the $c$-convolution is stable for this restricted set. In fact (132) at $c = 1$ is satisfied for log-concave distributions and since this set is stable by classical convolution, we have that the continuous probability distributions whose Markov-Krein transforms are log-concave is $c = 1$-stable.

- If one can find such a restricted set it will be interesting to know if it is possible to construct a random object (such as an infinite random matrix) associated to a measure belonging to this restricted set, together with a certain notion of *c-independence*, such that the $c$-convolution of measures would correspond to a sum of those $c$-independent random objects.

- Another interesting and open direction of research is to know if one can simplify the combinatorial formula of the moments-$c$-cumulants relations so that it can be expressed as a sum of $c$-weighted combinatorial objects, such as diagrams.

- In a previous note [50], the authors have introduced the multiplicative counterpart of the rank one HCIZ, whose asymptotic is governed by the logarithm the so-called $S$-transform of free probability (see also [41] for a similar rigorous derivation at $\beta = 2$). The formula in [50] suggests that we can operate a similar construction yielding a multiplicative $c$-convolution that interpolated between the classical multiplicative convolution and the free multiplicative convolution and we leave this problem for future work.

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
