# Peer review of "Rank one HCIZ at high temperature: interpolating between classical and free convolutions"

_SciPost Physics_

## Round 2 · Referee Report · Anonymous (Referee 1) · 2021-5-20

# Referee's report

## on **Rank one HCIZ at high temperature...**
by P. Mergny and M. Potters

This paper contains a novel and interesting discussion of an interpolation scheme between the classical and free convolutions. The authors introduce the so-called c-convolution, making use of the Marcus–Krein transform, and analyse in detail its properties. Many examples, involving quite elaborate analytical and numerical calculations are also provided. As such, this paper represents a real advance that might be useful in the context of Random Matrix Theory and its applications. But I regret that the authors didn't discuss in more detail the applications of this c-convolution that they have in mind.

The paper is, however, plagued with such a high number of typos, English mistakes, bad notations, defective punctuation, etc, that it makes the reader's (and the referee's) life quite miserable. It seems that some parts of the paper have been written too hastily and that one more reading and spell checking would have been useful.

Here is a list –presumably incomplete– of defects with my corrections or suggestions, with a couple of additional questions or remarks.

Throughout Section 1, the notation $\beta$ is used without any precise definition, until the beginning of sect. 2. Although the notation is familiar to any RMT aficionado, it would be better if it was recalled already in the first lines of Sect. 1.

Page 2, second paragraph: other types; there is only three ... $\to$ there are only three

three lines below: several attend $\to$ several attempts ?

three lines below: to cite few important results $\to$ to cite a few important results (a frequent mistake...)

two lines below: to not preserved $\to$ not to preserve

Third paragraph, line 6: the set of probability distributions

Lines 8 and 9 of second paragraph: derived $\to$ derive (another frequent mistake)

Section 2, title: Few words... A few words ...

Throughout the paper, the acronym HCIZ is used but, in my opinion, not really justified. As the authors know, Harish Chandra was dealing with integrals over orbits of the Lie algebra (Lie with a capital, see the first line of Sect. 2...). In the case $\beta = 2$ (the only case considered by Itzykson and Zuber), taking $A$ and $B$ Hermitian or anti-Hermitian makes no difference. But for $\beta = 1$ or 4, taking them real symmetric (or quaternionic self-dual) as in the present paper, or skew-symmetric (or quaternionic anti-self-dual), i.e., in the Lie algebra, makes a big difference. And this may be a source of confusion to the reader. The authors should either remove all references to HCIZ, or make this distinction clear and refer to generalized HCIZ, or angular integrals, or generalized or multivariate Bessel functions.

Page 4, line 3: to cite few recent ... $\to$ to cite a few recent

Two lines below (6), non-increasing integers

Four lines below (7), Specifying ? Do the authors mean Specializing ?

Page 6, below (18), maybe recalling what the Brézin–Hikami trick is would be helpful.

Page 7, below (27), $\sigma^2_{i,\beta} \to \sigma^2_{i,b}$

Page 8, in items 2. and 3. The summations look a bit awkward. Maybe $\sum_{\{(\pm)_i\}}$ would be better?

Bottom of that page, ... is known as a *mean...*: the authors should either give a reference or make the definition of that MK transform more explicit. The MK transform will be discussed at length later, but here, this is too elusive.

Top of page 9, the notations $a_{\min}$, $a_{\max}$ are defined as "the extreme values of [the vector] $\mathbf{a}$", which is not very precise. Their precise definition is given only below in section 2.3.1. The reader may guess what they are, but at this point, it looks strange.

Next line, the bounds ... is preserved. The authors should choose either the singular or the plural, but not both!

Next line, latter → later. What's the meaning of "relating the vector $\mathbf{a}$ to the distribution..." ?

Second line of eq. (39): $\mathcal{I}_{\mathbf{a}}^{(\beta)}(Nt)$

Page 10, third line: $zp \to tz$; $\qquad \mu_{\mathbf{a}} \to \mu_A$

Next line $p^*(z) \to z^*(t)$

Page 11, 5 lines below (51), "...its logarithm...": which logarithm?

Two lines below, "there is no interactions". Again, pick the singular or the plural, not both.

Question: Below (52), is there a simple argument why in that limit $\beta \to 0$, the averaging over $G^{(\beta)}$ reduces to one on the symmetric group $\mathrm{Sym}(N)$?

Page 15, first line of section 4. Why is the HCIZ integral now called HCIZ transform ?

End of that paragraph: an high temperature ... → a high temperature ...

Starting with eq. (80) and (82), the authors use previous notations for the functions $\mathrm{g}^{(\cdot)}$ and $U^{(\cdot)}$ with a different meaning of their subscript. To avoid confusion, I would suggest they introduce a small variant of their notation, like for example $\mathrm{g}^{(\beta)}$ and $\mathrm{g}^{[c]} := \mathrm{g}^{(\beta=2c/N)}$, and likewise for $U$.

Before (84), the Stieljes transform

Before (85), ... the usual Plemelj inversion ... to the inversion formula. This looks a bit heavy and redundant. Why not simply "One may recover the original distribution thanks to the usual Plemelj inversion formula (see (101)...):"?

Two lines below (85): we get that the following ... → we get the following

Same page, line -2: unbounded support,... from below, isn't it ?

One line below (88), converges

Two lines below: amounts to looking

Two lines before (93): It is worth ...

Before (96), observed → observe

At the end of eq (98), a semi-colon or a period, rather than a comma, would be more appropriate

One line below (106), is explained → are explained.

Three lines below (108) and the line after, functions, supports

Page 20, lines 4-5, for an original symmetric distribution $\mu$,

Next line, loss of generality

Two lines before sect 4.4: we give we give a few examples ... of distributions

In (118), what is the meaning of $\tilde{\beta}$?

Below (126), we have used that the imaginary part...

Two lines below, Stieltjes transform of $\nu$

First line of page 23, after (132), a colon or period would seem more appropriate than a comma, and five lines below, a comma rather than a semi-colon, after "in practice"

Two lines below (149), Stieltjes transform of the Bernoulli, and Bernoulli is spelled that way, with a single "i".

Sect. 5.1, line 4, equivalent to saying ...

Page 26, line 4, than → then; wrt → with respect to

Two lines below (153), All these properties ... Markov–Krein transform

Next line, the set of the images

Next line, what is meant by "is stable for probability measures" is presumably that the c-convolution of two probability measures is still a probability measure. But is even the positivity granted ? In view of the counter-examples of sect. 4.5, it is not clear. I see that the issue is mentioned in the Conclusion, but already here, this could be pointed out.

Next paragraph, We emphasize. Two lines below, be approximated. Two lines below, distributions . . . worlds

One line before (155), is given by. Two lines before (156), with get → we get. That paragraph, starting at "By the construction" and ending with the page, is too long, and I would suggest the authors cut it into several shorter sentences.

One line below (157), the first cumulant-moment . . .

Three lines below (158), we can obtain the moments

Four lines below (160) But we know from the previous . . .

Three lines below, interpolates → interpolate

Page 30, six lines below (162), any gamma (MP) distribution. Two lines below, Gaussian (semi-circle) random variable. . . gamma distribution

Next paragraph, what does "themselves" refer to?

Five lines below (165), both laws. Two lines below, $c$-gamma distribution . . . MP distributions

First line of page 31 should end up with a period rather than a comma: . . . is $c$-stable. Another...

Below (169), does not seemed → does not seem.

Before (172), Next using → Next use

Two lines below (175), approximated . . . results

In Fig. 5, the heavy dots of Poisson are hidden by the red curves

Page 34, line 2, we have constructed. Next line, convolutions. Below, HCIZ integral

Last paragraph, asymptotics. A punctuation seems to be missing between "logarithm" and "the so-called"

Line -2, that interpolates

Reference [14], the first author's name has been truncated

Reference [46], although Cauchy was not a very nice fellow, he deserves a capital. . .

Once these corrections and comments have been taken into account, the paper would be a good addition to SciPost.

---

## Round 2 · Referee Report · Anonymous (Referee 1) · 2021-5-20

Report

see below my report

Attachment

---

## Round 2 · Referee Report · Anonymous (Referee 2) · 2021-6-8

Strengths

1) Novel and interesting construction of a new convolution operation.

2) Creates new directions for future research.

Weaknesses

1) Poor presentation makes the paper unusually difficult to read.

Report

Please see attached pdf.

Attachment

  • validity: high
  • significance: high
  • originality: high
  • clarity: ok
  • formatting: below threshold
  • grammar: below threshold

---

## Editorial Decision

resubmitted